# Lipid-induced lysosomal damage after demyelination corrupts microglia protective function in lysosomal storage disorders

Enrique Gabandé-Rodríguez[1,2,*,‡], Azucena Pérez-Cañamás[1,†,‡], Beatriz Soto-Huelin[1], Daniel N Mitroi[1], Sara Sánchez-Redondo[3], Elena Martínez-Sáez[4], César Venero[5], Héctor Peinado[3,6] (ID) & María Dolores Ledesma[1,**] (ID)

## Abstract

Neuropathic lysosomal storage disorders (LSDs) present with activated pro-inflammatory microglia. However, anti-inflammatory treatment failed to improve disease pathology. We characterise the mechanisms underlying microglia activation in Niemann–Pick disease type A (NPA). We establish that an NPA patient and the acid sphingomyelinase knockout (ASMko) mouse model show amoeboid microglia in neurodegeneration-prone areas. *In vivo* microglia ablation worsens disease progression in ASMko mice. We demonstrate the coexistence of different microglia phenotypes in ASMko brains that produce cytokines or counteract neuronal death by clearing myelin debris. Overloading microglial lysosomes through myelin debris accumulation and sphingomyelin build-up induces lysosomal damage and cathepsin B extracellular release by lysosomal exocytosis. Inhibition of cathepsin B prevents neuronal death and behavioural anomalies in ASMko mice. Similar microglia phenotypes occur in a Niemann–Pick disease type C mouse model and patient. Our results show a protective function for microglia in LSDs and how this is corrupted by lipid lysosomal overload. Data indicate cathepsin B as a key molecule mediating neurodegeneration, opening research pathways for therapeutic targeting of LSDs and other demyelinating diseases.

**Keywords** cathepsin B; microglia; myelin; neurodegeneration; Niemann–Pick
**Subject Categories** Membrane & Intracellular Transport; Neuroscience
The EMBO Journal (2019) 38: e99553

See also: **A Silvin & F Ginhoux** (January 2019)

## Introduction

Lysosomal storage disorders (LSDs) comprise over 50 diseases that share similar features. Most of them have severe neurological components leading to cognitive and psychiatric alterations and early death. They are caused by mutations that induce the accumulation of subproducts from lysosomal degradation. For example, loss of function mutations in the gene encoding for acid sphingomyelinase (ASM), or for the cholesterol transport protein NPC1, causes Niemann–Pick disease type A (NPA) (Brady *et al*, 1966) or type C (NPC), respectively. Lysosomal lipid accumulation, demyelination and neurodegeneration characterise both diseases (Ledesma *et al*, 2011; Kodachi *et al*, 2017).

ASM catalyses the conversion of sphingomyelin (SM) into phosphorylcholine and ceramide in lysosomes (Stoffel, 1999). So, ASM-deficient cells accumulate high levels of SM in the lysosomal compartment and also in the plasma membrane (Galvan *et al*, 2008). Enzyme replacement therapy has proven successful to treat peripheral symptoms (Wasserstein *et al*, 2015). However, existing neurological deficits remain untreatable and constitute a major challenge to combat NPA. NPC1 is an endolysosomal membrane protein involved in intracellular cholesterol transport (Higgins *et al*, 1999). NPC1-deficient cells accumulate cholesterol (Vance & Karten, 2014) but also sphingomyelin (Devlin *et al*, 2010), gangliosides (Zervas *et al*, 2001) and sphingosine (Lloyd-Evans *et al*, 2008) in the endolysosomal compartment. Treatment with an inhibitor of glucosylceramide synthase that reduces ganglioside build-up (Zervas *et al*, 2001) stabilises neurological progression of NPC patients, but it does not cure the disease and has serious side effects. Lipid alterations in NPA and NPC diseases affect a number of biological processes including autophagy (Gabande-Rodriguez *et al*, 2014; Chung *et al*,

1  Department of Molecular Neuropathology, Centro de Biología Molecular "Severo Ochoa" (CSIC-UAM), Madrid, Spain
2  Barts Cancer Institute, Centre for Cancer & Inflammation, Queen Mary University of London, London, UK
3  Microenvironment and Metastasis Group, Molecular Oncology Program, Spanish National Cancer Research Centre (CNIO), Madrid, Spain
4  Department of Pathology, Hospital Universitario Vall d'Hebron, Barcelona, Spain
5  Department of Psychobiology, Universidad Nacional de Educación a Distancia, Madrid, Spain
6  Department of Pediatrics, Drukier Institute for Children's Health and Meyer Cancer Center, Weill Cornell Medical College, New York, NY, USA
   *Corresponding author. Tel: +34 91196 4535; E-mail: e.gabande@qmul.ac.uk
   **Corresponding author. Tel: +34 91196 4535; E-mail: dledesma@cbm.csic.es
   ‡These authors contributed equally to this work
   †Present address: Cellular Neuroscience, Neurodegeneration and Repair Program, Department of Neurology, Yale University School of Medicine, New Haven, CT, USA
   [The copyright line of this article was changed on 18 February 2019 after original online publication.]

2016), synaptic function (Karten *et al*, 2006; Camoletto *et al*, 2009; Hawes *et al*, 2010; Arroyo *et al*, 2014) and calcium homeostasis (Lloyd-Evans *et al*, 2008; Perez-Canamas *et al*, 2017).

An intriguing feature in NPA, NPC and other LSDs is the preferential and early death of Purkinje cells (PC) leading to motor impairments (Sarna *et al*, 2003). The specific cause of PC death, whether primary through intrinsic or extrinsic factors or secondary to the genetic defects, remains unknown and constitutes an intense, ongoing debate. Some studies suggest neurodegeneration causes microglia activation (Kollmann *et al*, 2012; Martins *et al*, 2015), while others propose it is a consequence of microgliosis caused by increased numbers of activated glia (Venkatachalam *et al*, 2008). This fundamental difference highlights the necessity to reveal the basic mechanisms that promote neuronal degeneration in LSDs.

Microglia are the resident macrophages of the central nervous system that have a unique origin deriving from primitive myeloid progenitors and are maintained independently of circulating monocytes throughout life (Ginhoux *et al*, 2010). As innate immune cells in the brain, activated microglia exert a number of functions such as migration, phagocytosis, antigen presentation and cytokine secretion (Nayak *et al*, 2014). Activated microglia occur in a pro-inflammatory phenotype that overexpresses markers such as inducible NOS, CD16/CD32 and MHC-II or in an anti-inflammatory pro-resolving phenotype that overexpresses CD206 and arginase-1 (Hu *et al*, 2015). Previous *in vitro* studies inducing pro- and anti-inflammatory differentiation suggest mutual exclusion of both types of microglia (Tang & Le, 2016). However, the *in vivo* scenario appears more complex, since certain neurodegenerative conditions contain a mix of both microglia populations and their intermediates (Miron *et al*, 2013). While microglia/macrophage-induced inflammation occurs in many LSDs (Wada *et al*, 2000), the therapeutic benefit of treatments targeting microglia-induced inflammation is limited (Vitner *et al*, 2012). These results suggested a secondary role for microglia in LSD pathology excluding these cells from further causal studies.

Microglia alterations could promote neuronal dysfunction in various neurodegenerative disorders (Perry *et al*, 2010; Bosch & Kielian, 2015). In contrast, microglia play beneficial roles in prion-induced neurodegeneration (Zhu *et al*, 2016) or amyotrophic lateral sclerosis (Spiller *et al*, 2018). A unique microglia type, the disease-associated microglia (DAM), has been recently associated with restricting development of Alzheimer's disease (Keren-Shaul *et al*, 2017). In-depth studies on microglia function in LSDs are limited and inconclusive. Here, we examined the role of microglia in NPA in detail using the ASMko mouse model and brain samples from an NPA patient. Our findings unexpectedly identify a subpopulation of microglia as a dominant factor that mediates NPA pathology, with a protective function that is corrupted after lysosomal damage. We find a similar microglia phenotype in the brain of an NPC patient and a mouse model of the disease, supporting these alterations as common pathological features in LSDs.

# Results

## Amoeboid microglia characterise brain areas with extensive neuronal death in a NPA patient and in ASMko mice

NPA neurodegeneration begins in cerebellar lobules, where PC suffer early death, and then extends to other brain areas during disease progression (Sarna *et al*, 2001). We wanted to determine whether microglia alterations occur and follow a distinct spatial pattern in NPA. We therefore analysed microglia throughout the brain of a NPA-affected individual. The low disease incidence and prognosis minimise accessibility to postmortem brain samples from NPA patients. We received neural tissue from a confirmed NPA-affected 3-year-old child, which we compared to an unaffected age-matched child. We analysed the expression of the microglia marker ionised calcium-binding adaptor molecule 1 (Iba-1). Compared to the control patient, Iba-1 immunostaining showed a substantial expansion of microglia in the cerebellum (Cb), cortex (Cx) and hippocampus (Hip) (Fig 1A) in the NPA patient. In a previous study, we showed extensive neuronal death occurs in these brain areas (Perez-Canamas *et al*, 2017). Virtually all microglial cells in the NPA patient presented an amoeboid morphology with increased cell area (Fig 1A–C), which is a characteristic feature of maximally activated microglia (Karperien *et al*, 2013).

The scarcity of brain samples from patients suffering this rare disease did not allow doing statistics for significance and moved us to analyse the ASMko mouse model for validation. As we observed in the NPA patient, analysis of microglia in symptomatic 3-month-old ASMko mice showed an increased number of these cells in the Cb (4.19-fold higher than in wt), Hip (3.61-fold) and Cx (2.02-fold) (Fig 1D and E). We also observed changes in microglia morphology in the Cb, where these cells exhibit round soma with drastically augmented area (517% with respect the wt) and almost no prolongations characteristic of amoeboid morphology (Fig 1D, F and G). Reduced length and branch points in microglia processes were significant in the Cx and Cb of ASMko mice (Fig 1H and I). Amoeboid morphology occurred exclusively in the Cb of ASMko mice, while in the Cx and Hip, microglia were enlarged but retained prolongations (Fig 1F). The drastic morphological change in the Cb, where neurodegeneration begins, suggests a link between fully activated amoeboid microglia and the onset of neurodegenerative processes.

## Microglial ablation worsens NPA phenotype in ASMko mice

Previous studies propose a toxic role for microglia in various LSDs. However, these studies only analysed cell morphology and levels of secreted cytokines and did not assess microglia directly. So, we analysed the specific contribution of microglia to the NPA phenotype. We treated ASMko mice with PLX5622 (PLX), a selective inhibitor of the colony-stimulating factor 1 receptor (CSF1R), which can completely ablate microglia in wild-type (wt) mice without inducing side effects (Dagher *et al*, 2015). To assess the contribution of microglia at early disease stages, we started the treatment at 1.5 months of age, when ASMko mice are not yet symptomatic (Macauley *et al*, 2008). We administered PLX in food pellets at a concentration of 290 mg/kg for 2 months. Ablation of microglia after the treatment was confirmed by Iba-1 immunostaining in the Cb of ASMko mice (Fig 2A–C).

We then examined motor performance in the Rotarod test, which confirmed motor impairment in vehicle-treated ASMko mice compared to vehicle-treated wt mice (Fig 2D and E). However, PLX-treated ASMko mice showed a significant decrease in the time spent on the rotating rod, indicative of worsened motor abilities, compared to vehicle-treated ASMko mice (Fig 2D and E). PLX

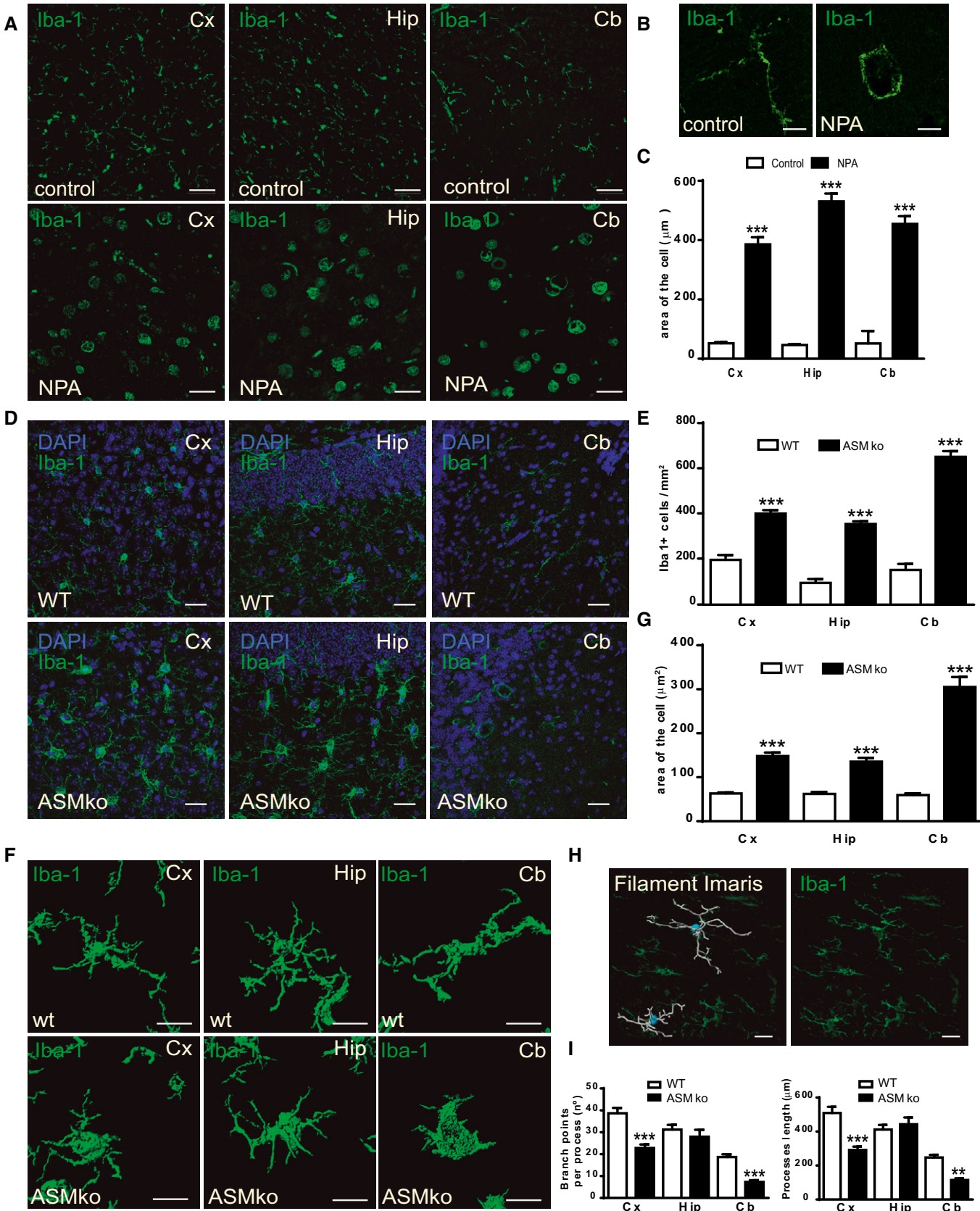

Figure 1.

**Figure 1.  ASM deficiency alters microglia number and morphology.**

A   Immunofluorescence staining against Iba-1 in cortex (Cx), hippocampus (Hip) and cerebellum (Cb) of NPA affected and control 3-year-old children. Scale bars, 50 μm.

B   Magnified image showing ramified (left) versus amoeboid (right) morphology in control and NPA microglia. Scale bars, 10 μm.

C   Mean ± SEM area of Iba-1-positive cells in different brain regions from a NPA affected and a control child (*n* = 20 cells, Student's *t*-test).

D   Immunofluorescence staining against Iba-1 in Cx, Hip and Cb of 3-month-old ASMko and wt mice. DAPI staining shows cell nuclei. Scale bars, 50 μm.

E   Mean ± SEM number of Iba-1-positive cells in different brain regions from ASMko and wt mice (*n* = 5 mice per group, Student's *t*-test).

F   3D rendered images obtained with Imaris Surface software from high magnification images of immunofluorescence staining against Iba-1 showing representative microglial morphology in Cx, Hip and Cb of ASMko and wt mice. Scale bars, 10 μm.

G   Mean ± SEM area of Iba-1-positive cells in the different brain regions from ASMko and wt mice (*n* = 5 mice per group, > 10 cells per mouse, Student's *t*-test). Scale bars, 15 μm.

H   Example of projection detection by the Filament Imaris software.

I    Mean ± SEM branch point number per process and process length in iba1[+] cells obtained with the Imaris Filament software (Student's *t*-test).

Data information: **P < 0.005; ***P < 0.001.

treatment did not have any deleterious effects on motor performance in wt mice (Fig 2D and E). These results prompted us to analyse the effect of PLX on ASMko mice life expectancy, which is typically 7–8 months (Horinouchi *et al*, 1995). We started long-term PLX treatment at 1.5 months until the end point. Analysis of body weight showed a significant decrease in PLX-treated ASMko mice compared to vehicle-treated ASMko mice (Fig 2F). Further, PLX treatment shortened life expectancy by 102 days (mean lifespan; ASMko veh: 269 days, ASMko PLX: 167 days) with 100% of PLX-treated ASMko mice dying before any vehicle-treated ASMko mice (Fig 2G). In contrast, neither survival nor body weight was affected by PLX treatment in wt mice (Fig 2F and G). To assess the role of microglia in PC death, we analysed the number of calbindin-positive cells after PLX short-term treatment. The number of calbindin-positive cells was significantly reduced in the middle zone (lobules VI, VII and VIII) and the posterior zone (lobules IX and X) of ASMko PLX-treated mice compared to vehicle-treated mice (Fig 2H and I). PLX treatment had no effects on PC from wt mice (Fig 2H and I). In contrast to previous proposals for a negative or secondary role for microglia in LSDs, our results suggest a protective and critical role of microglia on PC survival in NPA.

### Different microglia phenotypes coexist in the cerebellum of ASMko mice

It is generally accepted that microglia exist in a resting or active state. Active microglia can polarise in a continuum towards pro- and anti-inflammatory phenotypes (Hu *et al*, 2015). So, we characterised the specific microglia phenotypes in the Cb of ASMko mice.

We found the coexistence of the general microglia markers Iba-1 or F4/80 with the pro-inflammatory microglia marker CD16/CD32 or the anti-inflammatory microglia marker arginase-1 (Arg-1) (Fig 3A and B). Consistent with microglia-promoted inflammation, we detected increased mRNA expression and protein levels of the cytokine tumour necrosis factor alpha (TNFa) in the Cb of ASMko mice (Fig 3C and D). Further, induction of pro-inflammatory stimuli in the lipopolysaccharide (LPS) challenge by an injection of a sublethal dose of LPS led to 50% mortality in ASMko but not in wt mice. Co-administration of the anti-inflammatory corticosteroid dexamethasone completely reversed this effect (Fig 3E). These findings indicate functionally active pro-inflammatory microglia in ASMko mice consistent with the previous studies on other LSDs. However, this contradicts the above reported, deleterious effects found following general ablation of microglia suggesting pro-survival microglial activities.

### Arg-1-positive microglia phagocytose myelin debris in ASMko mice

Demyelination is found in many LSDs, including NPA (Buccinna *et al*, 2009; Platt, 2014). It has been shown that after demyelination processes (Neumann *et al*, 2009), microglia clear myelin debris, whose removal promotes remyelination to ensure neuronal survival (Lampron *et al*, 2015). Myelin-associated lipids such as SM promote phagocytosis and lipid catabolism in microglia (Turnbull *et al*, 2006; Poliani *et al*, 2015). We thus hypothesised that microglia could counteract demyelination in NPA. We confirmed myelin alterations in the Cb of ASMko mice using myelin basic protein (MBP)

**Figure 2.  Deleterious effects of microglia ablation in ASMko mice.**

A   Immunofluorescence staining against Iba-1 in Cb of ASMko and wt mice treated or not with PLX for 2 months. DAPI staining shows cell nuclei. Scale bar, 100 μm.

B   Magnified images (from A) showing ramified (left) versus amoeboid (right) morphology in wt and ASMko microglia treated or not with PLX. Scale bars, 30 μm.

C   Mean ± SEM number of Iba-1-positive cells in the Cb of the different mouse groups (*n* = 7 mice per group, two-way ANOVA, Bonferroni *post hoc*).

D, E  Performance in the Rotarod test of ASMko and wt mice treated or not with PLX for 2 months expressed as the mean ± SEM time spent in the rotating rod on each of four trials (D) or the mean ± SEM time of the four trials (E) (*n* = 7 mice per group, two-way ANOVA, Games–Howell (D) and Bonferroni (E) *post hoc*).

F   Mean ± SEM body weight of ASMko and wt mice treated or not with PLX at the indicated time during long-term treatment (*n* = 7 mice per group, two-way ANOVA, Bonferroni *post hoc*).

G   Survival curve of ASMko and wt mice with or without long-term PLX treatment (*n* = 7 mice per group).

H   Immunofluorescence staining against the PC marker calbindin in mid and posterior lobules of Cb in ASMko and wt mice treated or not with PLX for 2 months. DAPI staining shows cell nuclei. Scale bar, 500 μm.

I    Mean ± SEM number of calbindin-positive cells in the anterior (lobules I-V), middle (lobules VI-VIII) and posterior zone (lobules IX-X) of the Cb from the different mouse groups (*n* = 7 mice per group, two-way ANOVA, Bonferroni *post hoc*).

Data information: *P < 0.05; **P < 0.005; ***P < 0.001.

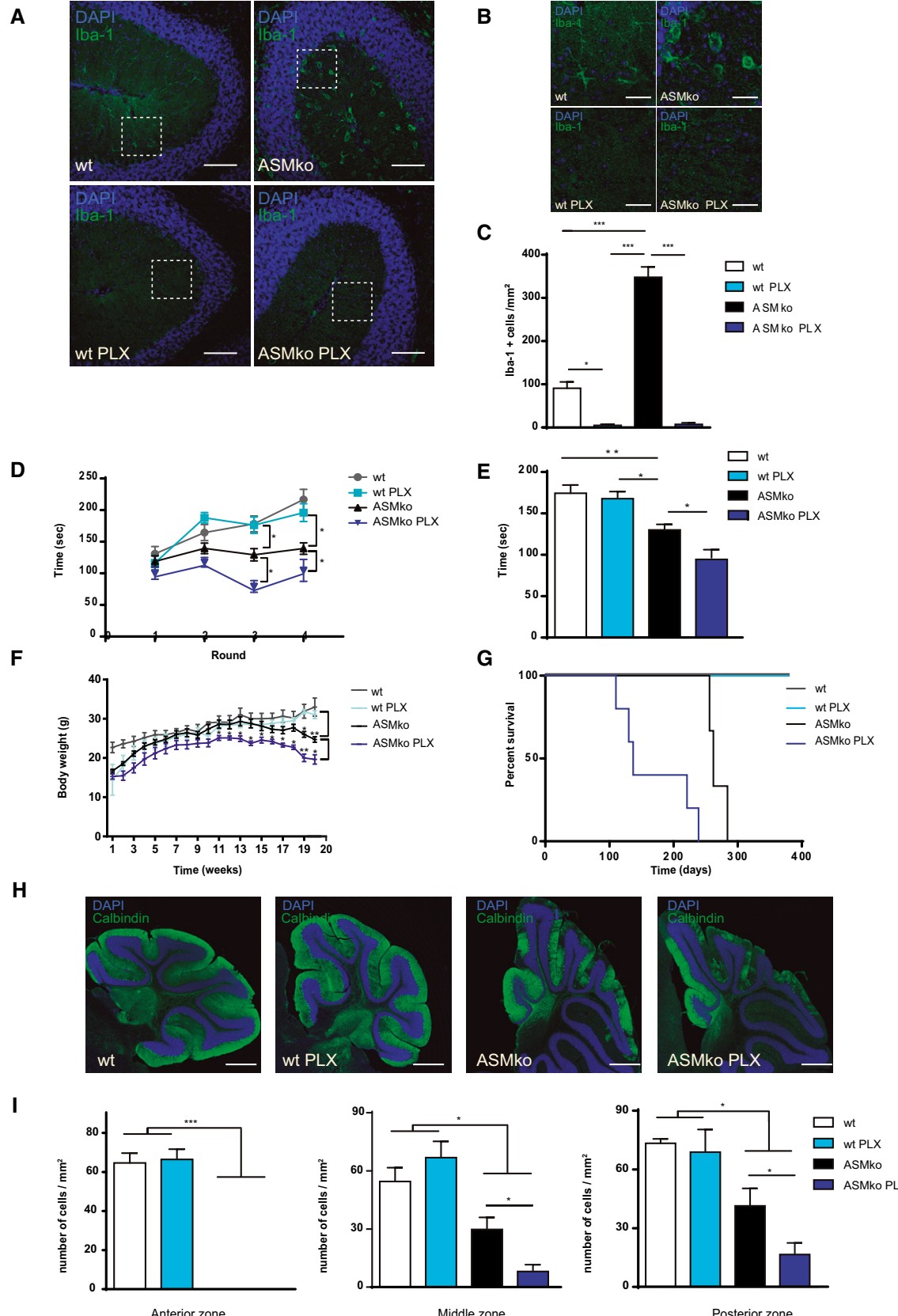

**Figure 2.**

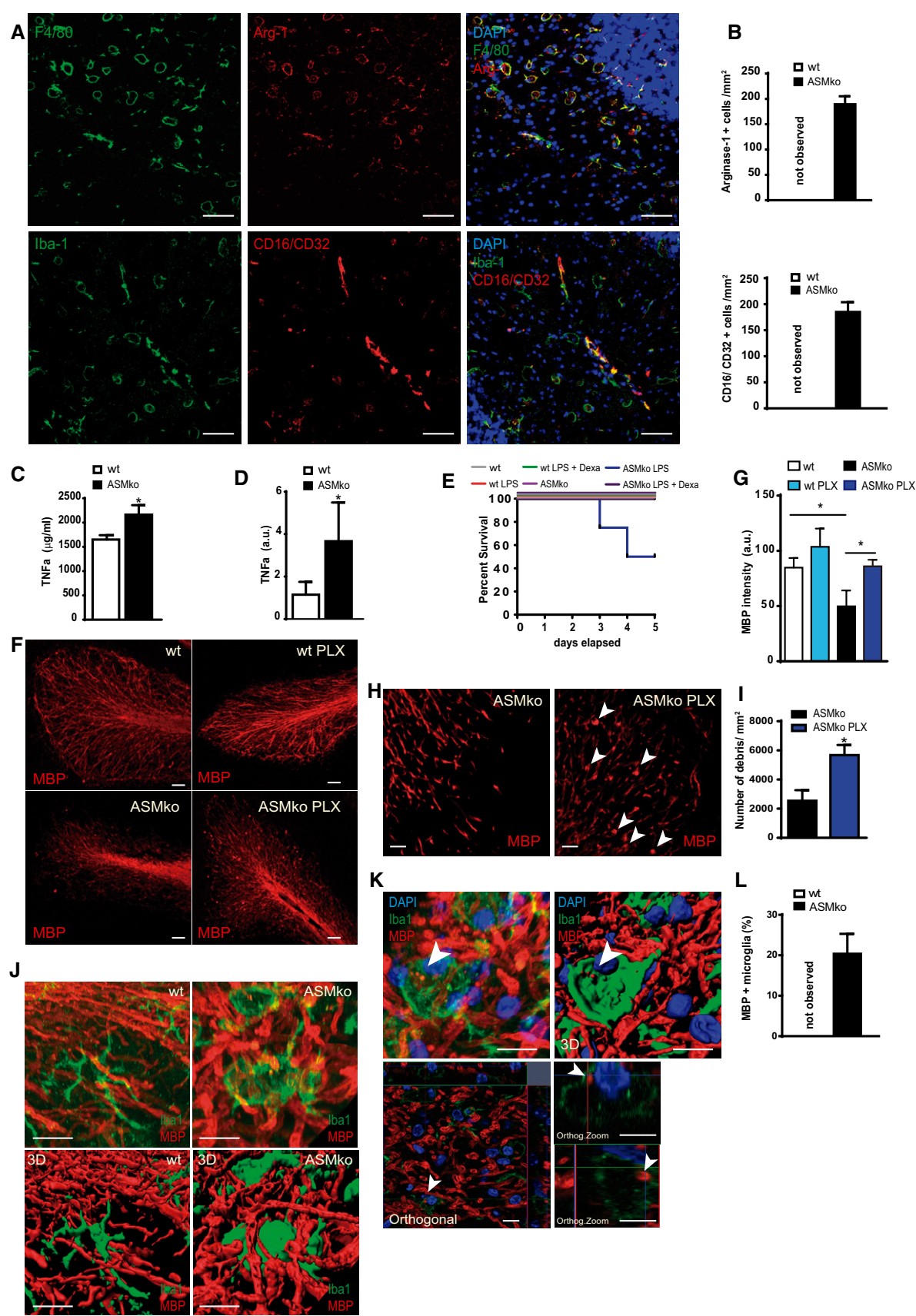

Figure 3.

**Figure 3.  Coexistence of different microglia populations and myelin debris in ASMko mice.**

A   Immunofluorescence staining against CD16/CD32 or Arg-1 in Cb of ASMko mice (red). Microglia were identified by F4/80 or Iba-1 staining (green). DAPI staining shows cell nuclei. Scale bar, 50 μm.

B   Mean ± SEM number of Arg-1 and CD16/CD32-positive microglia per area in Cb from ASMko and wt mice (*n* = 6 mice per group).

C   Mean ± SEM levels of TNFa protein in the Cb from ASMko and wt mice measured by ELISA (*n* = 7 mice per group, Student's *t*-test).

D   Mean ± SEM levels of TNFa mRNA in Cb of ASMko and wt mice determined by qRT–PCR (*n* = 7 mice per group, Student's *t*-test).

E   Survival after LPS challenge in ASMko and wt mice treated or not with dexamethasone (*n* = 8 mice per group).

F   Immunofluorescence staining against MBP in Cb of ASMko and wt mice treated or not with PLX for 2 months. Scale bar, 50 μm.

G   Mean ± SEM MBP intensity in the different mouse groups (*n* = 7 mice per group, two-way ANOVA, Bonferroni *post hoc*).

H   High magnification of immunofluorescence staining against MBP in ASMko mice treated or not with PLX for 2 months. White arrows show MBP aggregates indicative of myelin debris. Scale bar, 50 μm.

I   Mean ± SEM number of MBP aggregates per cell area in the different mouse groups (*n* = 7 mice per group, Student's *t*-test).

J   High magnification images and their 3D rendered replicates from cerebellar wt and ASMko microglia identified by Iba-1 and of the myelin marker MBP. Scale bars, 10 μm.

K   Amplified image and its 3D rendered replicate of amoeboid microglia containing myelin debris (arrowheads) in the ASMko cerebellum. The orthogonal projection of this cell in the lower panels confirms the presence of the myelin debris inside the cell and underneath the plasma membrane (arrowhead). DAPI staining shows cell nuclei. Scale bars, 10 μm.

L   Mean ± SEM percentage of microglia showing internalised myelin debris in wt and ASMko mice (*n* = 7 mice per group).

Data information: *P < 0.05.

immunostaining and found decreased staining intensity compared to wt mice (Fig 3F and G). Surprisingly, PLX-treated ASMko mice exhibited comparable MBP levels to wt mice (Fig 3F and G).

Altered myelination or myelin clearance could account for these differences in MBP content. So, we examined oligodendrocytes using the oligodendrocyte precursor marker Olig2 (Miron *et al*, 2011). We identified a similar number of oligodendrocytes in ASMko and wt Cb (Appendix Fig S1). Previous work had shown reduced mRNA and protein levels of myelin-specific proteins at postnatal stages, but not at birth, in ASMko mice (Buccinna *et al*, 2009) arguing in favour of impaired myelin maintenance rather than deficient oligodendrocyte generation and development. To gain insight into the myelination capacity of ASMko oligodendrocytes, we co-immunostained MBP and the axonal marker neurofilament 200. Consistent with impaired myelination, a higher percentage of axons (wt: 16.71%, ASMko: 32.89%) were not wrapped by MBP in the cerebellum of ASMko mice compared to wt (Appendix Fig S2). PLX treatment did not alter this pattern (Appendix Fig S2). However, we did notice an accumulation of rounded debris positive for MBP in PLX-treated ASMko mice not present in vehicle-treated ASMko mice or wt mice (Fig 3H and I). Moreover, ASMko microglia showed increased contact with MBP$^+$ axons compared to wt (Fig 3J). These observations supported altered microglial mechanisms to clear excess myelin debris in ASMko mice. Next, we quantified the number of Iba-1-positive cells that also expressed MBP-positive intracellular aggregates (Fig 3K and L). A significant 20.2% of microglia co-expressed intracellular myelin aggregates in ASMko mice, while we found no cells with internalised MBP aggregates in wt mice (Fig 3L). Further, virtually all microglia with positive intracellular MBP aggregates expressed Arg-1 (Appendix Fig S3). Altogether, these results indicate that a specific population of microglia counteracts demyelination during NPA by clearing myelin debris.

## Lysosomal damage and CathB release in ASMko microglia and macrophages

The above results suggest a beneficial role of microglia in NPA that could explain the disease worsening when microglia is ablated. Yet,

amoeboid microglia localise in brain areas with neurodegeneration (Fig 1). This prompted us to investigate the molecular mechanisms affecting microglia function in NPA. Accumulation of phagocytosed myelin in microglia can cause lysosomal lipofuscin-like inclusions and dysfunction in these cells (Safaiyan *et al*, 2016). We found abundant microgranular refringent deposits in Arg-1-positive microglia from ASMko mice (Appendix Fig S4). We identified these deposits as lipofuscin aggregates using autofluorescence quenching with Sudan black B (Appendix Fig S4). Given this finding and the SM-induced lysosomal membrane permeabilisation (LMP) we previously observed in ASMko neurons and reported in NPA fibroblasts (Gabande-Rodriguez *et al*, 2014), we hypothesised that lysosomal damage occurs in ASMko microglia. Transmission electron microscopy showed an accumulation of multilamellar bodies in microglia of ASMko mice (Fig 4A) similar to those found in ASMko neurons and NPA fibroblasts (Gabande-Rodriguez *et al*, 2014). To determine whether high SM levels induce LMP in microglia, we analysed the number of leaky vesicles transiently expressing Galectin-3-GFP (Gal3-GFP) in SM-treated BV2 cells, an inmortalised microglia cell line. Galectin-3 is normally a cytosolic protein recruited to damaged lysosomes and serves as a sensitive detector of lysosomal leakage (Maejima *et al*, 2013). We observed a 2.74-fold increase in the number of Gal3-GFP vesicles in SM-treated BV2 cells (Fig 4B and D). Addition of the ASM inhibitor, siramesine (Petersen *et al*, 2013), had a similar effect (Fig 4C and D). We also monitored for lysosomal leakiness in bone marrow-derived macrophages (BMDMs) and in postnatal cultured microglia from ASMko mice using a Lysotracker Red staining, which fluoresces in acidic environments (Wolfe *et al*, 2013). We used BMDMs as they share many characteristics with microglia and can be cultured from adult ASMko mice when SM accumulation is evident. On the other hand, SM increase in the microglia extracted from day 2 postnatal ASMko mice and maintained in culture for 14 days was confirmed by lysenin staining (Appendix Fig S5A and B). Lysotracker Red showed a diffuse pattern, indicative of lysosomal permeabilisation and cytosol acidification, in both ASMko BMDMs (Fig 4E) and postnatal microglia (Appendix Fig S5C).

These results suggest that SM overload can damage microglia/macrophage lysosomes. Cytosolic release of the lysosomal protease

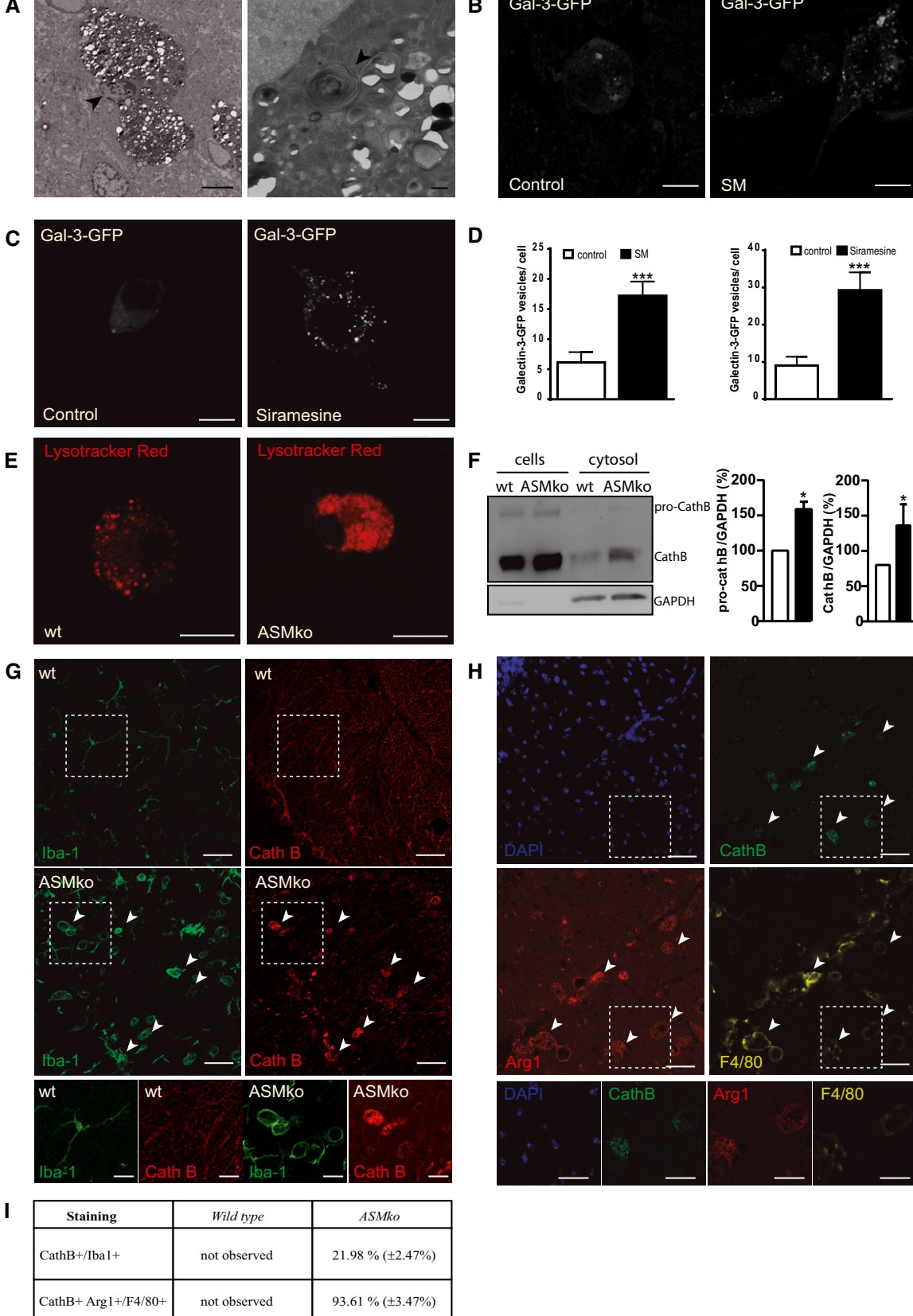

Figure 4.

**Figure 4.  SM mediated lysosomal damage and CathB accumulation in Arg-1-positive microglia of ASMko mice.**

A   Transmission electron microscopy image of microglia in the Cb of ASMko mice showing accumulation of multilamellar vacuoles and other lysosome-related vesicles filling the whole cytoplasm of these cells. Arrowhead identifies the nuclei. Scale bar, 5 μm. On the right, higher magnification showing a multilamellar body (arrowhead) with the characteristic concentric lamellar inclusions. Scale bar, 0.5 μm.

B, C   Immunofluorescence staining against Galectin-3-GFP in BV2 microglial cells treated with or without SM (B) or siramesine (C). Scale bar, 10 μm.

D   Mean ± SEM number of Galectin-3-positive vesicles in BV2 microglial cells in the different conditions ($n$ = 4 independent cultures, > 50 cells per culture, Student's $t$-test).

E   Lysotracker Red staining in BMDMs from wt and ASMko mice. Scale bar, 10 μm.

F   Western blot of CathB levels and the cytosolic protein GAPDH in the cellular and cytosolic fractions of wt and ASMko BMDMs extracted with digitonin (left). Mean ± SEM pro-CathB and CathB levels in the cytosolic fractions (right) ($n$ = 2 cultures per genotype, Student's $t$-test).

G   Immunofluorescence staining against Iba-1 (green) and CathB (red) in Cb of ASMko and wt mice. Arrowheads show double-positive cells. Scale bar, 50 μm. Below, magnified images of the selected areas. Scale bar, 20 μm.

H   Immunofluorescence staining against CathB (green), Arg-1 (red, anti-inflammatory microglia) and F4/80 (yellow, all microglia) in Cb of ASMko mice. DAPI staining shows cell nuclei. Scale bar, 50 μm. Below, magnified images of the selected areas. Scale bar, 20 μm.

I   Table showing mean ± SEM percentage of microglia presenting CathB staining (CathB[+]/Iba-1[+]) and the number of CathB-positive microglia co-expressing Arg-1 (CathB[+]/Arg-1[+]) in Cb of ASMko and wt mice ($n$ = 5 mice, > 10 cells per mouse).

Data information: *$P$ < 0.05; ***$P$ < 0.001.
Source data are available online for this figure.

CathB follows SM-induced LMP in ASM-deficient neurons and fibroblasts (Gabande-Rodriguez et al, 2014). We confirmed CathB cytosolic release in cultured BMDMs from ASMko mice by digitonin extraction (Fig 4F). *In vivo*, we found diffuse CathB staining in a significant percentage (21.98%) of Iba-1-positive cells in the Cb from ASMko but not wt mice, which further supports the occurrence of LMP in ASMko microglia (Fig 4G). SM is present in myelin (Di Biase et al, 1990), which is phagocytosed by Arg-1-positive microglia in ASMko brains (Appendix Fig S3). We hypothesised that this microglial subpopulation would be extremely sensitive to lysosomal damage. Indeed, 93.61% of microglia that expressed diffuse CathB staining were Arg-1-positive (Fig 4H and I).

Our results showing akin pathological phenotype in cultured ASMko BMDM and microglia, together with the amoeboid morphology of the iba-1 and F4/80-positive cells in the ASMko mouse brains, questioned whether infiltrated macrophages could be present. To answer this question, we performed costaining in the cerebellum of ASMko mice against F4/80 and a marker, TMEM119, which preferentially labels microglia but not macrophages (Bennett et al, 2016). A significant 62.37% of F4/80-positive cells in the ASMko Cb expressed low or no TMEM119 compared to wt (Appendix Fig S6). These results suggest that both microglia and infiltrated macrophages affected by lipid-induced lysosomal damage occur in the brains of ASMko mice. However, we cannot rule out that the F4/80-positive/low TMEM119 expressing cells are modified microglia similar to the recently described disease-associated microglia (DAM) in which the TMEM119 is downregulated (Keren-Shaul et al, 2017).

**ASMko-dependent extracellular CathB release mediates neuronal death**

Microglia/macrophage-secreted CathB can cause neurotoxicity *in vitro* (Kingham & Pocock, 2001) since this lysosomal protease retains activity in non-acidic environments (Boya & Kroemer, 2008). Paradoxically, extracellular CathB might also be neuroprotective under physiological conditions (Moon et al, 2016). The mechanisms involved in CathB secretion as well as whether it plays a role in neuronal death *in vivo* remain unknown. We thus analysed whether CathB was secreted in ASMko mice. Western blot analysis of CathB levels in the cultured media from ASMko and wt BMDMs (Fig 5A and B, Appendix Fig S11) or postnatal microglia (Appendix Fig S5D and E) showed increased levels of the cleaved and non-cleaved forms of CathB in ASMko conditions. So, we determined the mechanism by which CathB was released into the extracellular media after ASM deficiency using the more amenable BMDM cultures. Since activation of the ATP-gated receptor P2X7 promotes lysosomal leakiness and microvesicle-mediated CathB release from macrophages (Lopez-Castejon et al, 2010; Thomas & Salter, 2010), we used a specific P2X7 receptor antagonist (A740003) and measured the levels of CathB in the media of ASMko BMDMs. However, P2X7 inhibition slightly increased CathB release in ASMko macrophages, which argues against a role for the P2X7 channel in CathB release (Appendix Fig S7). Lysosomal insults can promote lysosomal protease secretion in several cell types via activation of $Ca^{2+}$-dependent exocytosis termed regulated lysosomal exocytosis (Olson & Joyce, 2015; Zhitomirsky & Assaraf, 2017). During this process, lysosomes fuse with the plasma membrane and release their content through lysosomal-associated membrane protein-1 (LAMP-1) (Yogalingam et al, 2008). We found a significant accumulation of LAMP-1 in microglia from ASMko, not wt, mouse brains (Fig 5C). This result suggests lysosomal exocytosis occurs in ASMko conditions. The 40.58% increase in plasma membrane-bound LAMP-1 in ASMko compared to wt non-permeabilised BMDMs further supports lysosomal exocytosis as a mechanism for CathB secretion upon ASM deficiency (Fig 5D and E). This is compatible with the appearance of both precursor and mature forms of CathB in the extracellular media since pro-CathB reaches the lysosome where it is processed by the action of proteases that require acidic pH (Reiser et al, 2010).

Recent evidence showed that extracellular secretion of lysosomal substrates accompanies the release of exosomes (Machado et al, 2015). To determine whether exosomes contributed to CathB release in ASMko conditions, we analysed whether CathB occurs in exosomes isolated from the conditioned media of ASMko and wt BMDMs. ASMko BMDMs secreted an increased number of exosomes, but not microvesicles (Fig 5F). We also confirmed this result by Western blot using the exosomal marker flotillin-1 (Flot-1) (Fig 5G). However, CathB was not detected in the exosomal fraction (Fig 5G), only in the exosome-depleted media (Fig 5H). These

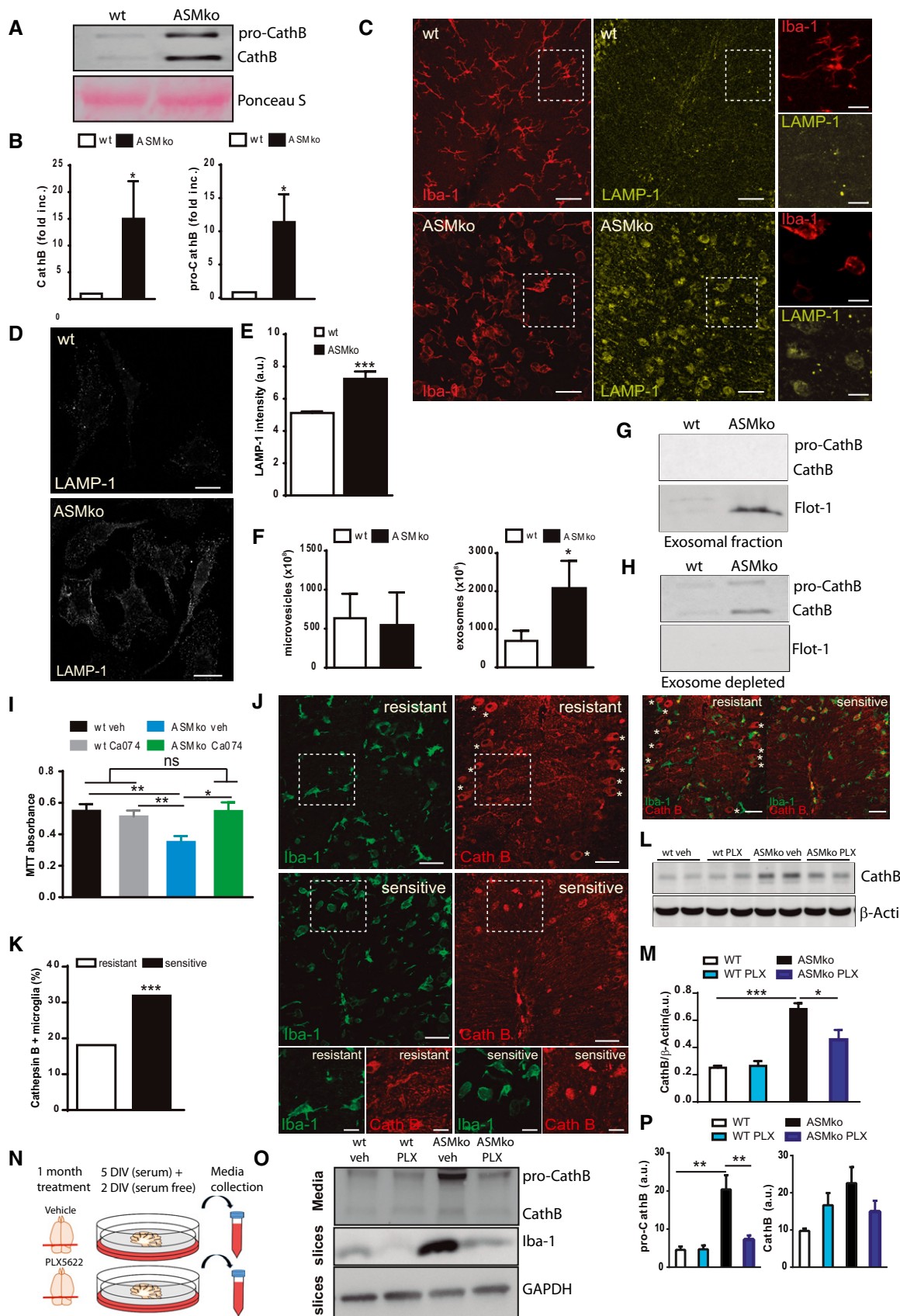

**Figure 5.**

**Figure 5.   Lysosomal exocytosis of CathB in ASMko conditions mediates neuronal death.**

A   Western blot of CathB levels (for both the precursor (pro-CathB) and the cleaved-mature forms) in the culture media from ASMko and wt BMDMs. Staining of Ponceau S is shown as loading control.

B   Mean ± SEM fold increase in CathB levels in the culture media from wt and ASMko BMDMs (*n* = 5 independent cultures, Student's *t*-test).

C   Immunofluorescence staining against LAMP-1 (yellow) and Iba-1 (red) in Cb of ASMko and wt mice. Scale bar, 50 μm, zoomed images 20 μm.

D   Immunofluorescence staining against LAMP-1 in non-permeabilised BMDMs from ASMko and wt mice. Scale bar, 50 μm.

E   Mean ± SEM of LAMP-1 levels on the surface of ASMko and wt BMDMs (*n* = 4 independent cultures, > 40 cells per culture, Student's *t*-test).

F   Mean ± SEM microvesicle and exosome numbers in the culture media from ASMko and wt BMDMs (*n* = 6 independent cultures, Student's *t*-test).

G   Western blot of CathB and flotillin-1 levels in the secreted exosome fractions from ASMko and wt BMDMs.

H   Western blot of CathB levels in the exosome-depleted culture media from wt and ASMko BMDMs.

I   Mean ± SEM MTT absorbance reflecting cell viability in primary neurons from wt mice treated with conditioned media from ASMko and wt BMDMs in the presence or absence of the CathB inhibitor Ca074 (*n* = 4 independent cultures two-way ANOVA, Bonferroni *post hoc*).

J   Immunofluorescence staining against Iba-1 (green) and CathB (red) in lobes of the Cb sensitive or resistant to Purkinje cell death. Asterisks indicate Purkinje cells in the resistant lobes. Scale bar, 50 μm.

K   Mean ± SEM percentage of CathB-positive microglia in resistant and sensitive Cb lobules of ASMko mice (*n* = 5 mice, chi-square test).

L   Western blot of CathB and β-Actin levels in cerebellar extracts of wt and ASMko mice treated or not with PLX.

M   Mean ± SEM of CathB levels normalised by β-Actin in cerebellar extracts of wt and ASMko mice treated or not with PLX (*n* = 6 mice two-way ANOVA, Bonferroni *post hoc*).

N   Diagram showing the experimental procedure to obtain media from cerebellar organotypic cultures dissected from the brain of wt and ASMko mice orally treated or not with PLX.

O   Western blot of CathB levels (for both the precursor (pro-CathB) and the cleaved mature forms) in the culture media from ASMko and wt cerebellar organotypic cultures from mice fed with vehicle (veh) or PLX (PLX). Below, Western blots of organotypic culture lysates against Iba-1, to confirm microglia depletion, and against GAPDH as loading control.

P   Mean ± SEM increase in pro-CathB (left) and CathB (right) levels in the culture media from cerebellar organotypic cultures from wt and ASMko mice treated or not with PLX (*n* = 3 mice per group two-way ANOVA, Bonferroni *post hoc*).

Data information: \*$P < 0.05$; \*\*$P < 0.005$; \*\*\*$P < 0.001$.
Source data are available online for this figure.

results indicate that CathB secretion by ASMko cells occurs through direct fusion of lysosomes with the plasma membrane rather than through exosome release.

Next, we tested whether secreted CathB could directly mediate neuronal death in ASMko conditions. Media collected from ASMko but not from wt BMDMs decreased the viability (36.06%) of wt neuronal cultures after a 24-h incubation as determined by the MTT assay (van Meerloo *et al*, 2011) (Fig 5I). Addition to the collected media of Ca074, a non-cell permeable inhibitor of CathB, reversed this phenotype and protected wt neurons from ASMko BMDM media-induced death (Fig 5I). Further, we found a striking correlation between CathB-positive microglia and PC death in the Cb of ASMko mice. PC death follows a currently unexplained pattern in ASMko mice. Some lobules retain most PC (resistant lobules), while others are preferentially affected during the neurodegenerative process (sensitive lobules) (Sarna *et al*, 2001). We analysed the number of CathB/Iba-1-positive microglia in PC-sensitive and PC-resistant lobules (Sarna *et al*, 2001). The analysis revealed an inverse correlation between the number of CathB-positive microglia and the number of PC (Fig 5J and K). This result suggests that CathB-positive microglia produce a specific susceptibility in PC to degenerate. Further *in vivo* evidences supported the role of microglia as CathB producing and secreting cells in ASMko mice. Thus, upon microglia ablation after 1-month-long oral treatment with PLX in ASMko and wt mice, we measured CathB levels in the cerebellar tissue (Fig 5L and M) and in the media of cerebellar organotypic cultures in which microglia depletion was confirmed by Iba-1 Western blot (Fig 5N, O and P, and Appendix Fig S11). We found increased Pro-CathB and a trend to increase CathB levels in the cerebellar tissue and in the secreted media of ASMko compared to wt mice. This was prevented by the microglia-depleting treatment with PLX.

**Pharmacological inhibition of CathB prevents NPA phenotype *in vivo***

Based on our results, we predicted that secreted CathB from microglia plays a key role in the Cb neurodegeneration found in ASMko mice. So, we assessed the effects of CathB inhibition *in vivo*. We implanted subcutaneously in 2-month-old ASMko and wt mice osmotic minipumps filled with the CathB inhibitor Ca074Me. The subcutaneously implanted pumps were connected to the brain via an intracerebral catheter to permit sustained drug release into the cerebrospinal fluid and to avoid the blood–brain barrier (Knafo *et al*, 2016). Ca074Me was administered for 1 month, and no changes in body weight in either group were found (Fig 6A). We analysed motor performance in the Rotarod test, which showed an expected decrease in time spent in the rod in ASMko compared to wt vehicle-treated mice (Fig 6B and C). ASMko mice treated with Ca074Me spent 36.79% more time on the rotating bar compared with vehicle-treated ASMko mice (Fig 6C). Ca074Me treatment did not show any effect in Rotarod performance in wt mice (Fig 6B and C). These results suggested a positive effect of CathB inhibition on PC survival. So, we quantified the number of calbindin-positive cells in different lobules of the Cb. Treatment with the CathB inhibitor, Ca074Me, did significantly increase the number of PC in Cb posterior lobules of ASMko mice (Fig 6D and E).

**Amoeboid morphology, myelin debris engulfment and CathB increase in mouse and human NPC1-deficient microglia**

Recent work has implicated CathB in cerebellar degeneration in NPC disease. Genetic ablation of the endogenous CathB inhibitor cystatin B dramatically enhanced PC death in an NPC1-deficient mouse model (Chung *et al*, 2016). This effect was considered cell

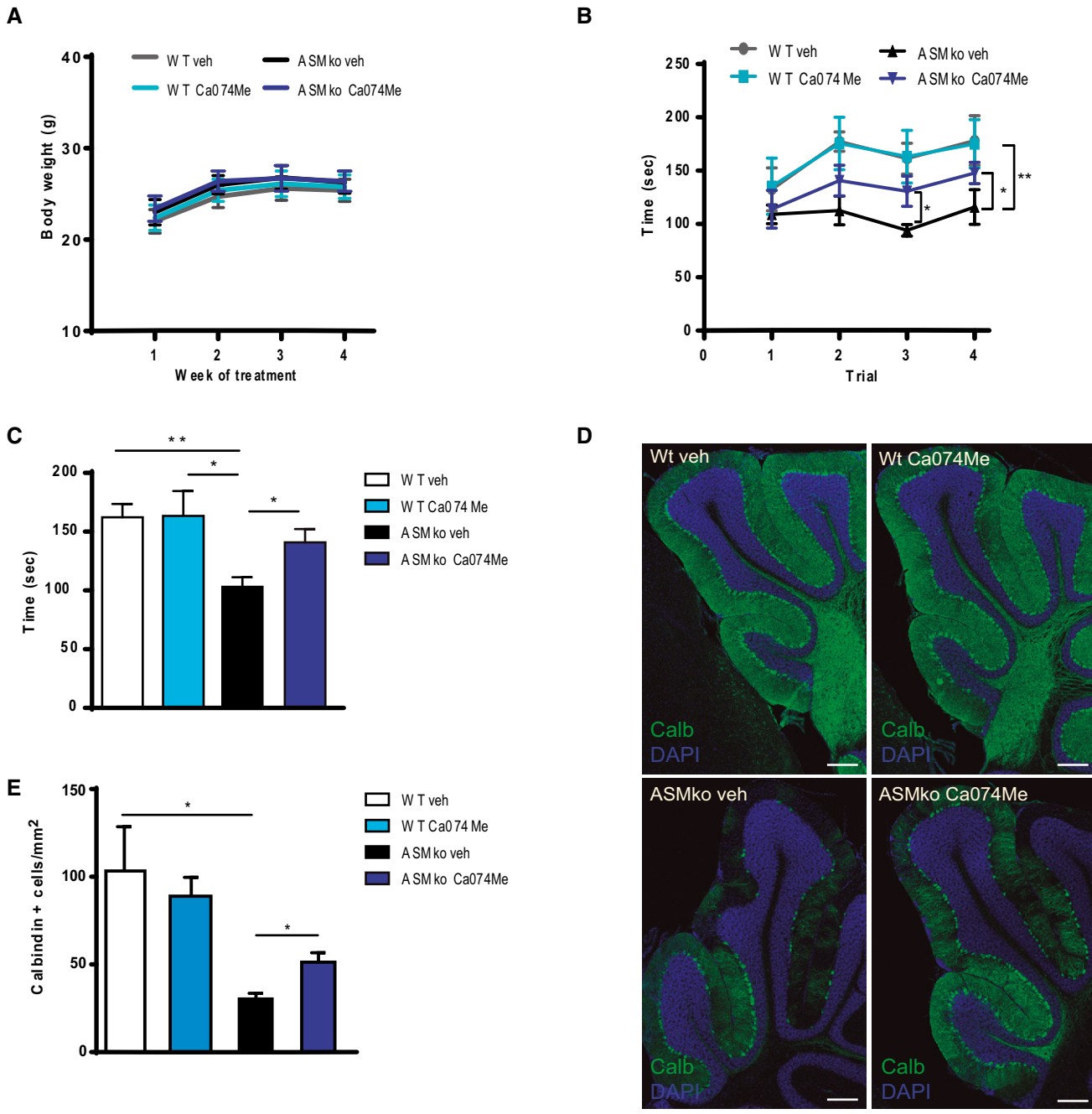

**Figure 6.  CathB inhibition prevents disease progression in ASMko mice.**

A    Mean ± SEM body weight of ASMko and wt mice treated or not with Ca074Me at the indicated time during treatment (*n* = 7 mice per group).

B, C    Performance in the Rotarod test of ASMko and wt mice treated or not with Ca074Me for 1 month expressed as mean ± SEM time spent in the rotating rod on each of four trials (B) or as mean ± SEM time of the four trials (C) (*n* = 7 mice per group two-way ANOVA, Bonferroni *post hoc*).

D    Immunofluorescence staining against the PC marker calbindin in the posterior lobules of the Cb in ASMko and wt mice treated or not with Ca074Me for 1 month. DAPI staining shows cell nuclei. Scale bar, 200 μm.

E    Mean ± SEM number of calbindin-positive cells in the posterior zone (lobules IX-X) of the Cb from the different mouse groups (*n* = 7 mice per group two-way ANOVA, Bonferroni *post hoc*).

Data information: **P* < 0.05; ***P* < 0.005.

autonomous. However, the contribution of microglia-derived CathB to PC degeneration observed in the mouse model for NPA led us to hypothesise a similar role in NPC. To test this possibility, we

analysed microglia in a mouse model for NPC (Npc1$^{nmf164}$) that bears a point mutation in the NPC1 gene highly similar to the most prevalent human mutation (Maue *et al*, 2012). Symptomatic

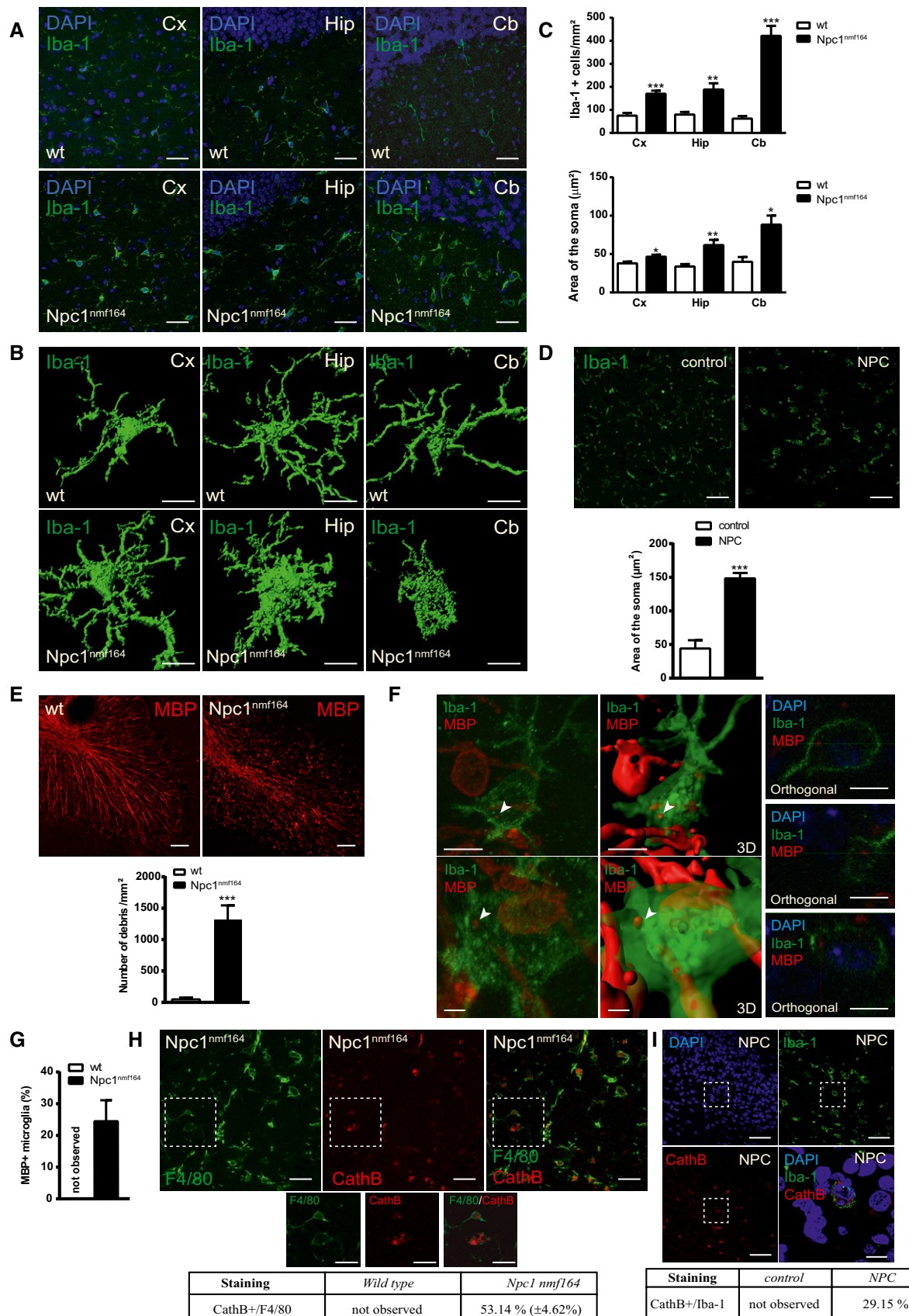

**Figure 7.**

◀

**Figure 7.  Alterations in NPC1-deficient microglia.**

A    Immunofluorescence staining against Iba-1 in Cx, Hip and Cb of 2-month-old Npc1[nmf164] and wt mice. DAPI staining shows cell nuclei. Scale bars, 30 μm.
B    3D rendered images obtained with Imaris Surface software from high magnification z-stack images of immunofluorescence staining against Iba-1 showing representative microglial morphology in Cx, Hip and Cb of Npc1[nmf164] and wt mice. Scale bars, 10 μm.
C    Mean ± SEM number (upper panel) and area (lower panel) of Iba-1-positive cells in different brain regions from Npc1[nmf164] and wt mice (n = 5 mice per group, > 30 cells per mouse, Student's t-test).
D    Immunofluorescence staining against Iba-1 in the Cb of a NPC-affected and a control child. Scale bars, 50 μm. Graph shows mean ± SEM area of Iba-1-positive cells (n = 30 cells, Student's t-test).
E    Immunofluorescence staining against MBP in Cb of Npc1[nmf164] and wt mice. Scale bar, 50 μm. Graph shows mean ± SEM number of MBP aggregates per area (n = 5 mice per group, Student's t-test).
F    High magnification images and their 3D rendered replicates from cerebellar Npc1[nmf164] microglia stained against Iba-1 and MBP containing myelin debris (arrowheads). Scale bars, 10 μm (up) and 2 μm (down). The orthogonal projection of the cells in the right panels confirms the presence of myelin debris inside the cell and underneath the plasma membrane. DAPI staining shows cell nuclei. Scale bars, 10 μm.
G    Mean ± SEM percentage of microglia showing internalised myelin debris in wt and Npc1[nmf164] mice (n = 5 mice per group).
H    Immunofluorescence staining against F4/80 and CathB in Cb of Npc1[nmf164] mice. Scale bar, 30 μm. The table shows mean ± SEM percentage of microglia presenting CathB staining in Npc1[nmf164] and wt mice. Below, magnified images of the selected areas. Scale bar, 20 μm.
I    Immunofluorescence staining against Iba-1 and CathB in Cb of a NPC patient. DAPI shows cell nuclei. Scale bar, 50 μm. A merged magnified image of the selected area is shown in the right down panel. Scale bar 10 μm. Table shows mean ± SEM percentage of microglia presenting CathB staining in NPC and control children.

Data information: *P < 0.05; **P < 0.005; ***P < 0.001.

2-month-old Npc1[nmf164] mice showed an increased number of microglia in the Cb (6.78-fold of wt), Hip (2.26-fold) and Cx (2.35-fold) (Fig 7A and C). We also observed changes in microglia morphology in the Cb, where these cells exhibit round soma with drastically augmented area (2.21-fold of wt) and almost no prolongation characteristic of amoeboid morphology (Fig 7A–C). Reduced length and branch points in microglia processes were significant in the Cx and Cb of Npc1[nmf164] mice (Fig 7B and C). Amoeboid morphology occurred exclusively in the Cb, while in the Cx and Hip, microglia were enlarged but retained prolongations (Fig 7B). These microglia alterations in the Npc1[nmf164] were remarkably similar to those observed in ASMko mice (Fig 1). As in the NPA patient, virtually all microglial cells in the Cb of an NPC patient presented an amoeboid morphology with increased cell area (Fig 7D), characteristic of maximally activated microglia (Karperien et al, 2013). Staining against MBP showed the accumulation of myelin debris in the Npc1[nmf164] mouse Cb compared to wt (Fig 7E). A significant 22.55% of Iba-1-positive cells contained MBP-positive intracellular aggregates (Fig 7F and G). In addition, consistent with the occurrence of LMP in Npc1[nmf164] microglia, we found increased diffuse CathB staining in a significant percentage (53.14%) of Iba-1-positive cells (Fig 7H). Remarkably, this staining was also evident in 29.15% of Iba-1-positive cells in the NPC patient (Fig 7I).

## Discussion

Here, we provide a detailed characterisation of microglia in NPA and NPC, to elucidate the specific role of these cells in *in vivo* LSD models and to clarify a long-standing debate on their role in these diseases. Increased activated microglia and elevated levels of pro-inflammatory cytokines occur in different LSDs (Jeyakumar et al, 2003; Vitner et al, 2012; Xiong & Kielian, 2013; Martins et al, 2015; Groh et al, 2016). However, anti-inflammatory drugs and genetic ablation of pro-inflammatory cytokines show no or minimal benefits to disease progression (Pedchenko et al, 2000; Smith et al, 2009; Vitner et al, 2012, 2014). These findings suggest no role or a secondary contribution of microglia to LSD pathology. This paradox has remained unresolved given the lack of detailed microglia characterisation. Our results provide

evidence for an unexpected dual role and for a major pathological impact of microglia in LSD brain pathology.

The polarisation process of microglia into pro- and anti-inflammatory phenotypes typically yields two mutually exclusive populations. However, we show that microglia expressing markers for both types coexist in the brain of the mouse model for NPA, quintessential LSD. On one hand, microglia in ASMko mice boost the levels of TNFa, which are also increased in other LSDs (Vitner et al, 2012). In agreement with an enhanced basal inflammatory state supported by microglia, ASMko mice were vulnerable to septic shock. While this result could suggest a negative role for these cells and could explain the high susceptibility to infections in NPA patients (McGovern et al, 2006), our experiments using PLX-induced microglia ablation uncovered an unexpected protective role. While potential cytotoxic cell debris released by microglia ablation might contribute to worsen the disease after PLX treatment, the lack of toxicity observed in PLX-treated wt mice (our results and Dagher et al, 2015) would argue against this possibility. We rather propose that Arg-1-positive microglia can clear myelin debris that accumulates during disease progression and therefore microglia elimination would be deleterious. Yet, lipid-induced overloading of Arg-1-positive microglia lysosomal capacity disrupts their beneficial function to produce lysosomal damage and exocytosis and the extracellular release of CathB, inducing neurotoxicity (see model in Fig 8). To our knowledge, this is the first time that a shift from a protective to a deleterious role of microglia is described in a neurodegenerative disease. Since demyelination and lysosomal lipid accumulation occur in many LSDs (Platt, 2014), we speculate this mechanism may be conserved. Supporting this view are the strikingly similar alterations of microglia we observed in an NPC patient and Npc1[nmf164] mouse model. During the preparation of this manuscript (Cougnoux et al, 2018, 2018) described a similar microglia phenotype in the Npc1[−/−] mouse, which mimics the most aggressive forms of NPC. Also supportive are the reports showing that microglial activation precedes neuronal death in Sandhoff disease (Wada et al, 2000); defective clearance capacity of glia cells promotes neurodegeneration in a *Drosophila* model of mucolipidosis type IV (Venkatachalam et al, 2008); and bone marrow transplants with non-mutated cells that transmigrate into the brain and produce microglia-like cells are protective in mouse models of

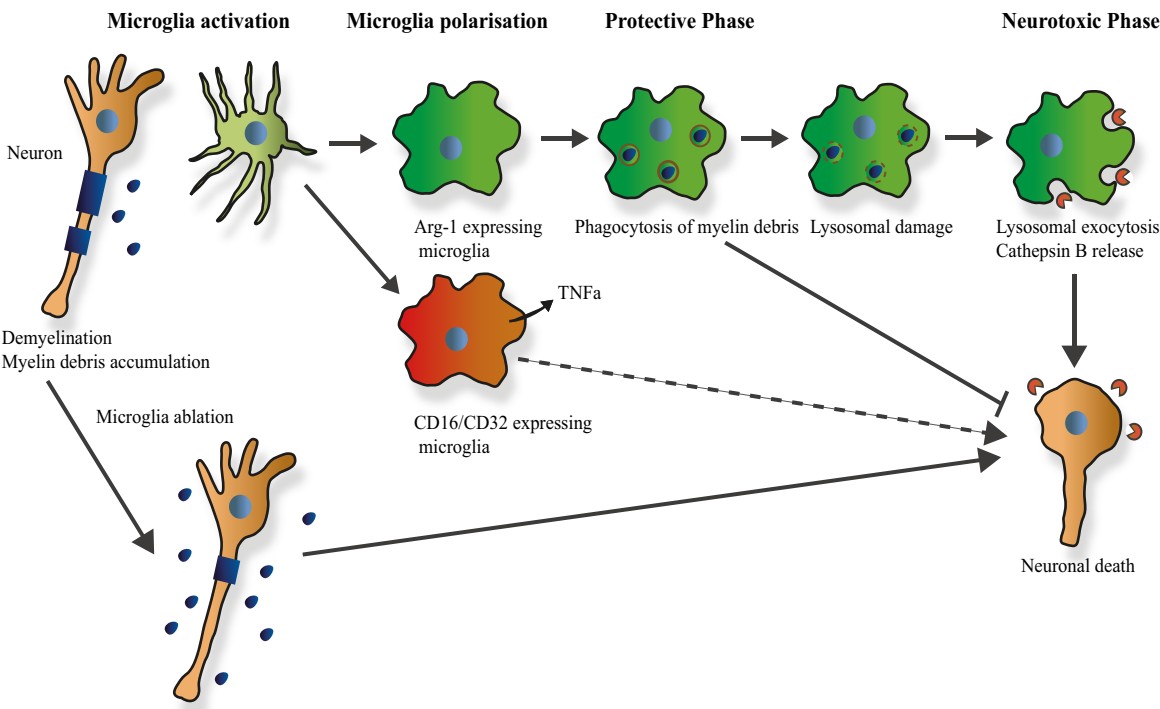

**Figure 8.  Model for microglia contribution to LSD pathology.**

Initial stages of LSD pathology show direct lipid storage-induced neuronal degeneration, demyelination and myelin debris (shown in dark blue). These events activate resting microglia, which polarise into CD16/CD32 and Arg-1 expressing phenotypes. General microglia ablation causes myelin debris accumulation, which enhances neuronal death. CD16/CD32 expressing microglia promote increased levels of cytokines such as TNFa, which could amplify neuronal death. Arg-1 expressing microglia phagocytose and remove myelin debris protecting neurons from their toxic effects (protective phase). Continued lipid overload of lysosomes, due to the genetic defects and myelin debris accumulation, causes lysosomal damage and activates lysosomal exocytosis in microglia. This process stimulates extracellular release of CathB (shown in red), which directly causes neuronal death (neurotoxic phase).

mucolipidosis type IV (Walker & Montell, 2016), mucopolysaccharidosis type I (Pievani *et al*, 2015), Fabry disease (Yokoi *et al*, 2011), Sandhoff disease (Wada *et al*, 2000) and NPC (Lee *et al*, 2010).

Microglia cells arise from a unique type of embryonic precursors (Ginhoux *et al*, 2010). However, in certain neurological diseases, the brain myeloid cell compartment can also be composed by infiltrating monocytes from peripheral blood. Distinguishing both populations is not trivial since infiltrating monocytes differentiate into microglia-like cells under the brain milieu (Capotondo *et al*, 2017). The transmembrane receptor TMEM119 has been proposed as a specific microglia marker that is absent in blood-derived macrophages (Bennett *et al*, 2016). Our finding that a percentage of F4/80-positive cells express no or low levels of this marker supports the presence of infiltrated macrophages in the ASMko mice. However, a recent study uncovered a new subtype of DAM protective microglia in mouse models for Alzheimer's disease and amyotrophic lateral sclerosis (Keren-Shaul *et al*, 2017) in which TMEM119 expression is downregulated. In contrast, CathB is upregulated in these cells and may contribute to the phagocytosis of undegraded material. Whether the myeloid cells that do not express TMEM119 or express it at low levels in ASMko brains are DAM, monocyte-derived "microglia-like" cells or both remains to be clarified. Very recent reports suggest the absence of monocyte infiltration in the brain of mouse models for LSDs including NPC (Cho *et al*, 2018; Cougnoux *et al*, 2018). These findings would support that the TMEM119-negative cells are DAM. Irrespective of the origin, we predict that

lipid-induced lysosomal damage and exocytosis in these CathB-enriched cells will stimulate the extracellular release of this protease contributing to neurotoxicity as disease progresses.

Our results suggest that microglial activation occurs via intrinsic (due to the genetic alteration in these cells) and extrinsic (secondary to alterations in other cells) factors in LSDs. In NPA, we show that SM accumulation due to ASM deficiency can cause lysosomal damage, induce microglia to release CathB and trigger neuronal death. At the same time, engulfment of myelin further increases SM levels and lysosomal damage in microglia to perpetuate this cycle. Cholesterol is a major lipid component of myelin (Saher *et al*, 2005) and the primary lipid accumulating in NPC due to NPC1 deficiency. Although studies reporting a stabilising role of cholesterol in lysosomal membrane (Appelqvist *et al*, 2011) would argue against this lipid-promoting LMP in NPC microglia, recent work has described that cholesterol-rich myelin debris can promote lysosomal membrane rupture in aged phagocytes of a mouse model for demyelination (Cantuti-Castelvetri *et al*, 2018). Given the remarkable SM increase that accompanies cholesterol alterations in the lysosomes of NPC1-deficient cells (Devlin *et al*, 2010), we speculate that the cooperative action of both lipids may enhance microglia corruption in NPC.

Cathepsins have been related to several neurodegenerative disorders. However, the underlying cellular and molecular mechanisms and their pathological relevance have not been elucidated. Moreover, recent studies suggest beneficial roles of circulating CathB in neurogenesis and memory function (Moon *et al*, 2016). Intracellular

lysosome-to-cytosol translocation of CathB has been described in cellular models of NPC (Chung *et al*, 2016) and of NPA (Gabande-Rodriguez *et al*, 2014). Here, we demonstrate the key neurotoxic role of amoeboid microglia-released extracellular CathB after lysosome overload that may underlie the striking and unexplained patterned cerebellar PC death in many LSDs. Our results show a strong correlation between CathB-positive amoeboid microglia and cerebellar lobules of ASMko mice in which PCs were preferentially lost. This result argues for a role for amoeboid microglia to establish this pattern. While this type of cells is restricted to certain Cb lobules in ASMko mice, they do occur in other areas like the Cx and Hip in the NPA patient brain. This may explain why there is extensive neuronal death and very fast progression in the human disease, while neuronal death is only detected in the Cb and lifespan is relatively longer in the mouse model. Noteworthy, increased CathB staining accompanies the presence of amoeboid microglia in the NPA patient (Appendix Fig S8) and, evidencing the impact of this results in other LSDs, the same is observed in a mouse model and a patient suffering NPC. While neuronal death after lysosomal leakage has been observed before in LSDs (Gabande-Rodriguez *et al*, 2014), microglial death has not been reported and is unlikely to happen as the number of microglia is consistently increased in LSDs (Wada *et al*, 2000; Vitner *et al*, 2012; Martins *et al*, 2015). By cleaved caspase 3 staining and TUNEL assays, we confirmed that microglia do not undergo apoptosis in the mouse model for NPA neither (Appendix Fig S9). We propose that microglia triggers lysosomal exocytosis of CathB to avoid LMP-induced apoptosis but ends up causing neuronal death as a side effect, significantly contributing to foster neurodegeneration. CathB could cause neurotoxicity indirectly by prompting pyroptosis (Brojatsch *et al*, 2015) and/or activating the inflammasome pathway (Duewell *et al* 2010) in microglia. However, the prevention of neuronal death we achieved by addition of the non-cell permeable CathB inhibitor to the media collected from ASMko macrophages, and not to the macrophages themselves, argues in favour of a direct neurotoxic effect of the secreted CathB. On the other hand, the reduced levels we found in the ASMko mouse brains of interleukin 1b (IL-1b), which is a main component of pyroptosis and the inflammasome pathways (Appendix Fig S10), would also argue against the toxic contribution of these mechanisms.

Our results provide critical insight into an intensely debated feature of LSDs and stimulate innovative pharmacological therapies. An advantage of CathB as a specific therapeutic option is that several inhibitors are currently available for experimental purposes (Siklos *et al*, 2015). Clinically, novel CathB inhibitors with a higher half-life that penetrate the blood–brain barrier will provide additional therapeutic options to prevent the neurological symptoms in LSDs.

# Materials and Methods

### Antibodies

Antibodies against the following proteins or epitopes were used for Western blots and immunofluorescence: Iba-1 (rabbit polyclonal, Wako, 019-19741), calbindin D-28k (mouse monoclonal, Swant, 300), F4/80 (rat monoclonal, Abcam, ab6640), Arg-1 (goat polyclonal, Santa Cruz, SC-18354), CD16/CD32 (rat monoclonal, BD Pharmingen, 553141), MBP (rat monoclonal, AbD Serotec,

MCA409S), CathB (goat polyclonal, Santa Cruz, SC-6493), CathB (rabbit polyclonal, Chemicon Int, AB4064), CathB CA10 (mouse monoclonal, Abcam, ab58802), CathB (C-19) (rabbit polyclonal, Santa Cruz, SC-6490), CathB (goat polyclonal, R&D Systems, AF965), LAMP1 (rat monoclonal, DSHB, 1D4B), flotillin-1 (mouse monoclonal, BD Transduction Laboratories, 610821) and Olig2 (rabbit polyclonal, Millipore, AB9610), cleaved caspase 3 (Asp 175) (rabbit polyclonal, Cell Signaling, 9661), TMEM119 (rabbit polyclonal, Abcam, ab209064), NF200 (mouse monoclonal clone NE14, Sigma-Aldrich, N5389), GAPDH (mouse monoclonal, Abcam, ab8245) and b-Actin (mouse monoclonal, Sigma-Aldrich, A2228). HRP-conjugated goat anti-rabbit and rabbit anti-mouse antibodies (DakoCytomation) were used as secondary antibodies.

### TNFa ELISA

TNFa levels were determined by ELISA in total cerebellar extracts according to manufacturer instructions (mouse TNF-alpha Quantikine ELISA Kit, R&D systems).

### Human samples

Formaldehyde-fixed brain tissue from a 3-year-old NPA patient, a 14-day-old NPC patient and a 3-year-old control child was donated by the Wylder Nation Foundation (http://wyldernation.org), the Vall d'Hebron Hospital and the Fundación CIEN brain bank (http://bt.fundacioncien.es), respectively.

### Mice

Breeding colonies were established from ASM heterozygous C57BL/6 mice (Horinouchi *et al*, 1995), kindly donated by Prof. EH Schuchman (Mount Sinai School of Medicine, New York, NY, USA) and from Npc1[nmf164] mice (Maue *et al*, 2012) purchased from Jackson laboratories. Male and female ASMko, Npc1[nmf164] and wild-type (wt) littermates were analysed at 3 and 2 months of age, respectively. No gender-dependent differences were observed in any of the results. Procedures followed European Union guidelines and were approved by the CBMSO Animal Welfare Committee.

### Immunofluorescence

Mouse brains were dissected, fixed and cryoprotected. The tissue was then frozen in Tissue-Tek optimal cutting temperature compound (Sakura Finetek, Torrance, CA, USA), and 30-μm sagittal sections were obtained with a cryostat (CM 1950 Ag Protect freezing: Leica, Solms, Germany). The sections were incubated overnight at 4°C with the primary antibodies and then with the corresponding Alexa-conjugated secondary antibodies (Invitrogen). Finally, the sections were incubated for 10 min with DAPI (Calbiochem), washed and mounted with Prolong Gold Antifade (Invitrogen). Images were obtained on a confocal LSM710 microscope (Carl Zeiss) and quantified using the Fiji software (Schindelin *et al*, 2012). The percentage of demyelinated axons was calculated using the Metamorph 7.10.0.119 software. 3D rendered images from z-stacks and analysis of microglia processes were obtained with Imaris Surface or Filament tools, respectively (Bitplane). Orthogonal projections of microglia engulfing myelin debris were obtained with Zen

Digital Imaging Software (Zeiss). For TMEM119 quantification, an average threshold of intensity was established in wt sections and cells below that threshold were considered low-expressing or negative cells.

Human sections were deparaffinised in decreasing concentrations of ethanol and xylene and subjected to heat-mediated antigen retrieval in citrate buffer (pH 5.9). Afterwards, the same protocol performed for mouse sections was carried out. Finally, sections were dehydrated by immersion in increasing concentrations of ethanol and xylene and mounted using FluorSave (Merck Millipore).

## Immunohistochemistry

Human slices were deparaffinised and rehydrated by immersion in decreasing concentrations of ethanol and xylene. Immediately, heat-mediated antigen retrieval was performed in antigen unmasking solution (Vector Laboratories) for 15 min. After incubation with a biotinylated secondary antibody (Vector Laboratories), avidin–biotin–enzyme–elite peroxidase-based solution was applied according to manufacturer instructions (ABC Kit, Vectastain Laboratories). Development was achieved by incubating the sections with 3-3′-Diaminobenzidine (Sigma-Aldrich). Finally, nuclei were counterstained with Hematoxylin Gill solution (Sigma-Aldrich) and sections were dehydrated as above and mounted in DPX (Sigma-Aldrich). Images were taken in a Zeiss Axiophot microscope (Carl Zeiss).

## PLX5622 treatment

PLX5622 was provided by Plexxikon Inc. and formulated in AIN-76A standard chow by Research Diets Inc. PLX5622 was provided *ad libitum* at 290 mg/kg.

## Electron microscopy

Mice were transcardially perfused with PBS and fixative (4% paraformaldehyde and 2% glutaraldehyde in PBS). Brains were postfixed in 4% PFA overnight and sectioned in 200-μm-thick slices. Cerebellar sections were embedded in Epon, stained with uranyl acetate and lead citrate, and examined with a transmission electron microscope (JEM1010, Jeol, Akishima, Tokyo, Japan). Microglial cells identified were photographed with a CMOS 4k TemCam-F416 camera (TVIPS, Gauting, Germany).

## Lipofuscin autofluorescence and Sudan Black B staining

Autofluorescence associated with lipofuscin aggregates was detected in brain sections at excitation and emission wavelengths of 360 and 540–640 nm, respectively. Incubation for 10 min with Sudan black B (10 mg/ml: Santa Cruz, Dallas, TX, USA), which quenches lipofuscin autofluorescence, confirmed this pigment identity.

## Quantitative RT–PCR

Total RNA from wt and ASMko cerebellum was extracted with TRIzol Reagent (Ambion/RNA Life Technologies Co.) following the manufacturer instructions using the RNeasy Mini Kit (Qiagen, Hilden, Germany). RNA was quantified by absorbance at 260 nm using a NanoDrop ND-100 (Thermo scientific; Themo Fisher Scientific Inc.). Retrotranscription to first-strand cDNA was performed using RevertAid H Minus First-Strand cDNA Synthesis Kit (Thermo scientific; Thermo Fisher Scientific Inc.). Briefly, 5 ng of synthesised cDNA was used to perform fast qPCR using GoTaq qPCR Master Mix (Promega Co., Madison, WI, USA) in ABI PRISM 7900HT SDS (Applied Biosystems; Life Technologies Co.) following manufacturer instructions. The primers, purchased from Sigma-Aldrich (mouse TNFa: forward; 5′-AGGGATGAGAAGTTCCCAAA-3′ and reverse; 5′-TGGGCCATAGAACTGATGAGA-3′), were used at 0.5 μM final concentration. Three housekeeping genes (Gapdh, GusB and Pgk1) were used as endogenous controls.

## LPS sublethal challenge

ASMko and wt mice were i.p. injected with 1 mg/kg LPS *E. coli* O111:B4 (Calbiochem). When indicated, 2 mg/kg of the anti-inflammatory corticosteroid dexamethasone (Sigma-Aldrich) was co-injected. This LPS dose causes upregulation of central cytokines and sickness behaviour with 100% of wt mice surviving (Silverman *et al*, 2013). Survival was monitored over time.

## Galectin-3-GFP transfection in BV2 cells

The immortalised mouse microglial cell line BV2 was obtained from the American Type Culture Collection and cultured with DMEM containing 10% FBS. Galectin-3-GFP plasmid was a kind gift from Prof. Tamotsu Yoshimori (Maejima *et al*, 2013). BV2 cells were plated in glass coverslips and transfected using Lipofectamine 2000 (Life Technologies) according to the manufacturer's instructions. 48 h after transfection, the cells were treated with SM (40 μM, Sigma-Aldrich) or with the ASM inhibitor siramesine (30 μM, Sigma-Aldrich) during 24 h. Images were taken on a confocal LSM510 microscope (Carl Zeiss).

## IL-1b milliplex assay

IL-1b was determined in cerebellar extracts from 3-month-old wt and ASMko mice using the Milliplex technology (MCYTOMAG-70K-PMX, Millipore) according to manufacturer instructions.

## BMDM cultures

The femur and tibia from 4-month-old ASMko and wt mice were harvested and washed with PBS. Bone marrow-derived cells were flushed out with DMEM containing 10% foetal bovine serum (FBS). The cell suspension was filtered through a 70-μm cell strainer to remove any cell clumps. The single-cell suspension was then cultured in 80% DMEM medium containing 10% FBS and M-CSF (20% L929-conditioned medium was used as a source of M-CSF) for 8 days to promote monocyte differentiation into macrophages. The generated BMDMs were F4/80$^+$ (purity > 90%).

## Neuronal cultures

Primary cultures of hippocampal neurons were prepared from wt day 18 embryos as described (Kaech & Banker, 2006). Neurons were kept under 5% CO$_2$ at 37°C in Neurobasal medium (Gibco) plus B27 supplement (Gibco) and GlutaMAX (Gibco) until 8 days *in vitro*

(DIV). Then, the medium was replaced with Neurobasal medium plus B27 without GlutaMAX.

### Postnatal microglia cultures

Microglia were obtained from P2 mice as previously described (Mecha *et al*, 2011). 800,000 cells were kept in DMEM media without serum for 3 days after which media was collected and concentrated as explained for BMDM cultures.

### Cerebellar organotypic cultures

Cerebellar slices were prepared from 3.5-month-old wt and ASMko mice either treated with vehicle or PLX5622 for 1 month as explained in the PLX5622 treatment. Cerebella were extracted in dissection solution (10 mM D-glucose, 4 mM KCl, 26 mM NaHCO$_3$, 233.7 mM sucrose, 5 mM MgCl$_2$ and 1:1,000 phenol red), oxygen saturated with carbogen (95% O$_2$/5% CO$_2$) and sliced in an automatic tissue chopper (McIlwain Tissue Chopper, Standard Table, 220 V, Ted Pella Inc.) to obtain 350-μm slices. The slices were placed in porous membranes and kept at 35.5°C and 5.5% CO$_2$ for 5 days in media containing MEM, 20% horse serum, 1 mM glutamine, 1 mM CaCl$_2$, 2 mM MgSO$_4$, 1 mg/ml insulin, 0.0012% ascorbic acid, 30 mM Hepes, 13 mM D-glucose and 5.2 mM NaHCO$_3$. Osmolarity of the media was adjusted to 320. At day 5, horse serum was removed and replaced by serum-free media that was collected at day 7 and concentrated as explained for BMDM cultures. The total amount of protein in the slices from every animal was obtained by the bicinchoninic acid assay (BCA, Pierce) and was used to normalise the amount of media. Slices and media were further analysed by Western blot against CathB, Iba-1 and GAPDH.

### Lysotracker staining

Lysotracker Red DND99 (Thermo Fisher Scientific) was added to ASMko and wt BMDMs or primary cultured microglia at 1 μM concentration and incubated for 15 min. Cells were then fixed in 4% PFA and mounted using Prolong Gold Antifade (Invitrogen). Images were taken on a confocal LSM710 microscope (Carl Zeiss).

### CathB measurement in BMDM media

For CathB analysis in the media, 4 × 10$^5$ BMDMs were plated in 12-well plates with DMEM media. 24 h-conditioned media was removed and centrifuged at 900 *g* for 5 min to eliminate any cell contamination and cell debris and concentrated using 3 kDa Amicon Ultra-4 Centrifugal Filter Units (Merck Millipore) before performing Western blot. Ponceau S (Sigma) was used as a loading control for CathB quantification. The P2X7 receptor inhibitor A740003 (50 μM, Tocris Bioscience) was added to BMDMs when indicated and incubated for 24 h before collecting the conditioned media. CathB levels in the media were analysed by Western blot as previously described (Perez-Canamas *et al*, 2016).

### Exosome purification, characterisation and analyses

Exosomes were purified by differential centrifugation as previously described (Peinado *et al*, 2012; Hoshino *et al*, 2015). Briefly, cells were removed from the cell culture supernatant by centrifugation at 500 *g* for 10 min to remove any cell contamination. To remove any possible apoptotic bodies and large cell debris, supernatants were then spun at 12,000 *g* for 20 min. Exosomes were harvested by spinning at 100,000 *g* for 70 min, washed in 7 ml of PBS and pelleted again by ultracentrifugation (Beckman 70.1Ti rotor). Exosome preparations were verified by electron microscopy. Exosome size and particle number were analysed using the NS500 nanoparticle characterisation system (NanoSight) equipped with a blue laser (405 nm). The final exosome pellet was resuspended in PBS, and protein concentration was measured by the bicinchoninic acid assay (BCA, Pierce).

### LAMP1 surface staining in BMDMs

ASMko and wt BMDMs were washed with PBS and fixed with 2% paraformaldehyde in PBS and 0.12 M sucrose for 8 min. Immunofluorescence was performed using a conventional protocol in the absence of detergent to avoid cell permeabilisation.

### Lysenin staining

Lysenin staining in primary microglia was performed as previously described (Gabande-Rodriguez *et al*, 2014).

### Digitonin cytosolic extraction

For cytosol extraction, cultured BMDMs were incubated for 15 min in Hepes buffer pH 7.5 containing 15 μg/ml of Digitonin (Sigma-Aldrich) as described in Gabande-Rodriguez *et al* (2014).

### TUNEL assay

For apoptosis analysis, 30-μM-thick PFA-fixed cerebellar sections were stained with the click-iT Plus TUNEL assay (Thermo Scientific, C10617) according to manufacturer instructions. Sections were imaged in a LSM800 (Carl Zeiss).

### Cell viability assay

Cell viability was evaluated using the MTT (methylthiazolyldiphenyl-tetrazolium bromide, Sigma) assay. Briefly, 3 × 10$^4$ neurons were plated in 96-well plates coated with poly-L-lysine (0.1 mg/ml). At DIV12, serum-free 48 h-conditioned media from ASMko or wt BMDMs was added to neuronal media at a 1:1 ratio in the presence or absence of the non-cell permeable CathB inhibitor Ca074 (50 μM, Sigma-Aldrich). After 24 h, 1 mM MTT was added to the wells and incubated for 3 h. Then, the supernatant was discarded, and 50 μl DMSO was added to the plates. The colour intensity was measured at 570 nm using a microplate reader (FLUOstar Optima, BMG Labtech). Cells treated with DMSO were used as a control.

### Inhibition of CathB *in vivo* with osmotic minipumps

ASMko and wt mice were anesthetised with isoflurane, and i.c.v. delivery cannulas (brain alzet kit III) were implanted with a stereotaxic frame (KOPF Instruments) at the following coordinates according to the bregma: AP, −0.5 mm; ML, 1 mm; and DV, −2.2 mm.

Osmotic minipumps (Alzet; model #1004) were filled with Ca074Me (Sigma-Aldrich) or vehicle (0.9% NaCl, 1.5% DMSO), equilibrated in 0.9% NaCl at 37°C for 48 h, attached to the i.c.v. cannula tubing and subcutaneously implanted at the back. Ca074Me was administered at 1 mg/ml at a rate of 0.25 μl/h as previously described (Hook *et al*, 2008). Animals were housed individually. One month after implantation, behavioural testing was performed and then mice were sacrificed for histological analyses.

### Behavioural analysis

The Rotarod test was performed in an accelerating Rotarod apparatus (Ugo Basile, Varese, Italy). Mice were trained for 2 days at a constant speed: the first day—four times at 4 r.p.m. for 1 min and on the second day—four times at 8 r.p.m. for 1 min. On the third day, the Rotarod was set to progressively accelerate from 4 to 40 r.p.m for 5 min. Mice were tested four times. During the accelerating trials, the latency to fall from the rod was measured.

### Statistics

Data from at least three different experimental groups were quantified and presented as the mean ± SEM. Normality of the data was tested using the Shapiro–Wilk test. For two-group comparisons, the Mann–Whitney *U*-test for non-parametric data or a two-sample Student's *t*-test for data with parametric distribution was used. For multiple comparisons, data with a normal distribution were analysed by two-way ANOVA followed by Bonferroni or Games–Howell post hoc test. The statistical significance of non-parametric data was determined by the Kruskal–Wallis test to analyse all experimental groups. The Mann–Whitney *U*-test was used to analyse paired genotypes, applying the Bonferroni correction. $P < 0.05$ were considered significant. In the figures, asterisks indicate the *P*-values: *< 0.05; **< 0.005; ***< 0.001. SPSS 23.0 software (IBM, Armonk, NY, USA) was used for all statistical analyses.

**Expanded View** for this article is available online.

### Acknowledgements

We thank Steven and Shannon Laffoon at the Wylder Nation Foundation and Dr. A. Rábano at the Fundación Cien for donating brain samples. We also thank the confocal, electron microscopy and graphic design services of CBMSO, Plexxikon Inc., for providing PLX5622, Prof. J.J. Lucas from CBMSO for sharing behavioural test devices, Dr. Oscar Rogero from Merck for advice and technical assistance with the Milliplex assay, Dr. Fernando de Castro from Cajal Institute for sharing reagents and advice and Dr. Sarah Spear for language correction. This work was financed by grants from the Ministerio Economía y Competitividad (SAF2014-57539-R and SAF2017-87698-R) and the Fundación Niemann Pick España to MDL, and by an institutional grant to the CBMSO from the Fundación Ramón Areces. EG-R and AP-C held fellowships from Fundación Niemann Pick España and the Ministerio Español de Ciencia e Innovación (FPU), respectively. We thank Life Science Editors for editing services.

### Author contributions

EG-R, AP-C and MDL designed the experiments and wrote the paper. EG-R, AP-C BS-H and DNM performed the experiments. EM-S collected and prepared human samples. CV made the surgery for osmotic minipump implantation. SS-R and HP designed and performed exosome studies.

### Conflict of interest

The authors declare that they have no conflict of interest.

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
