## [Review Process File · The EMBO Journal]

Lipid induced lysosomal damage after demyelination corrupts microglia protective function in Lysosomal Storage Disorders

Enrique Gabandé-Rodríguez, Azucena Pérez-Cañamás, Beatriz Soto-Huelin, Daniel N. Mitroi, Sara Sánchez-Redondo, Elena Martínez-Sáez, César Venero, Héctor Peinado and María Dolores Ledesma.

Review timeline:

Submission date:	3 rd April 2018
Editorial Decision:	17 th May 2018
Revision received:	7 th September 2018
Editorial Decision:	5 th October 2018
Revision received:	12 th October 2018
Accepted:	25 th October 2018

Editor: Elisabetta Argenzio

Transaction Report:

1st Editorial Decision

17th May 2018

Thank you for submitting your manuscript (EMBOJ-2018-99553) on the protective roles of microglia in lysosomal storage disorders to The EMBO Journal. We have now received three referee reports on your study, which are enclosed below for your information.

As you can see, while the referees judge the findings to be overall important and interesting, they also raise critical points that need to be addressed before they can support publication at The EMBO Journal. In particular, referee #1 is concerned that the study suffers from misinterpretation of your results on neurotoxicity of CathepsinB release and suggests expanding the characterization of pro-versus anti-inflammatory microglia. In addition, referee #3 points to concerns regarding causalities between demyelination and microglia, as well as the physiological relevance of the findings obtained with BMDM cells.

Addressing these issues as suggested by the referees would be essential to warrant publication in The EMBO Journal. Given the overall interest of your study, I would thus like to invite you to revise the manuscript in response to the referee reports.

REFeree REPORTS

Referee #1:

The authors study the contribution of microglia in the pathogenesis of the lysosomal storage disorder caused by deficiency of acid sphingomyelinase (ASM) in mice. ASM KO mice are characterized by an increase in microglia number and activation status. Ablation of microglia using

the compound PLX5622 worsen the disease as measured by behavioral, lethality and morphological analyses. Using markers against Arginase1 and CD16 or F4/80, the authors find activated microglia polarized towards both "M1" and "M2" phenotypes. The Arginase1 positive microglia are the cells that primarily engulf myelin debris. A fraction of these cells exhibited signs of lysosomal damage as shown by CathepsinB release. Medium transfer experiments show that microglia releasing CathepsinB exert neurotoxic effects, which was rescued by a CathepsinB inhibitor both in vitro and in vivo. Finally, the authors extend their findings to NPC, a second lysosomal storage disease. In summary, this paper enhances our understanding of lysosomal storage diseases and provides a novel therapeutic strategy. The experiments are convincing and of high quality. Overall, the paper is of great interest and I therefore recommend publication after considering the following points:

My main criticisms concern the interpretation of some of the results. I am not fully convinced that the release of CathepsinB from microglia is directly neurotoxic. There is evidence that lysosomal damage leads to release of CathepsinB into the cytosol of phagocytes, where it can activate the inflammasome pathway (Düwell et al. Nature 2010), which in turn could mediate the neurotoxic effects for example via IL1b. The authors should discuss this possibility in the discussion.

The authors classify the microglia into pro- and anti-inflammatory cells based on single markers. If the authors want to keep this nomenclature more work is necessary. The authors would need to use additional markers and in particular markers that indicate pro- and anti-inflammatory changes.

Alternatively, the cells could be isolated and analyzed by RNA-Seq. However, as these pro- and anti-inflammatory pathways are not part of this study, I recommend avoiding these labels.

The authors refer to sphingomyelin as a lipid enriched in myelin. In spite of its name, sphingomyelin is depleted from myelin.

Page9: "We found diffuse CathB staining in a significant percentage of Iba1 positive cells in the Cb from ASMko but not wt mice, which supports LMP in ASMko microglia (Figure 4 F, H)".

Additional assays are necessary to prove this point. Increased labeling is easy to see, but the diffuse labeling is not sufficient to prove lysosomal membrane rupture. The authors could analyze lysosomal pH or determine biochemically the cytosolic fraction of CathepsinB or use other assays. The authors should in addition, test whether microglia become TUNEL positive. Release of Cathepsin into the cytosol is known to trigger pyroptosis.

The authors ablate microglia and show that this leads to worsening of the disease. Ablation of cells leads to release of potentially cytotoxic cell debris. This possibility should be discussed as a reason for the deterioration of the disease.

Referee #2:

In this article the authors studied the role of microglia in the evolution of neuropathic Lysosomal storage disorders (LSDs) as Nieman Pick disease. In order to elucidate the role of microglia in the development of this disease the authors studied patient samples and a sphingomyelinase knockout mouse model (ASMko). By studying the morphology of the cells among different brain regions the authors showed that microglia acquires an amoeboid shape due to their activation. The most unexpected result is that the depletion of microglia increases the pathological symptoms in the KO mouse model, suggesting a neuroprotective contribution of the microglia. However the authors were able to show the existence of two microglia phenotypes based on Cathepsin B (CathB) and Lamp1 expression. Revealing the presence of microglia phagocytosing large amount of myelin debris inducing an overloading of the lysosomes and their exocytosis to the extracellular space, the main consequence being the release of CathB playing a neurotoxic role in the evolution of the disease. The authors succeeded to show that inhibiting CathB activity in this mouse model reduces neuronal death and behavioral anomalies.

The article is well written and clear. All the information necessary to understand and follow are cited. The experiments are well designed and controls are there. All the results are presented in a very clear manner and not over interpreted.

I have only few comments:

- In the introduction the author should refer to the uniqueness of microglia origin (Ginhoux et al 2010).
- Following on this, how the authors can formally exclude that the cells that they are looking are monocyte-derived cells that colonize the brain and not embryonic derived microglia that could die

due to the KO.

- In the discussion the authors could discuss about the recent description of the disease-associated microglia, also called DAM, performed by Keren-Shaul et al 2017 and how this could be related to the microglia they described in this article.

Referee #3:

EMBOJ-2018-99553, corr. author Dr. Ledesma

"Lipid induced lysosomal damage after demyelination corrupts microglia protective function in Lysosomal Storage Disorders"

Reviewer comment:

The manuscript 'Lipid induced lysosomal damage after demyelination corrupts microglia protective function in Lysosomal Storage Disorder' nicely demonstrates both the protective as well as the detrimental effect of microglia in Niemann Pick disease. Although it was already known that microglia phagocytose myelin debris thereby promoting remyelination, Gabandé-Rodríguez et al. show here for the first time with different in vitro and in vivo models and in a human patient that microglial Cathepsin B release from the lysosomes is contributing to neuronal degradation in NPA and NPC. Furthermore, they demonstrate that inhibition of Cathepsin B release ameliorates disease symptoms like Purkinje cell death and motor deficiencies.

The manuscript is well structured and written and the topic - dissecting the effect of pro- and anti-inflammatory microglia populations and their functions in LSD - is certainly of high interest. However, there are a couple of major concerns, which should be addressed:

Major points.

1. Supplementary Figure S1: To exclusively link the demyelination process to the microglia, one would need to not only count the number of oligodendrocytes, but rather assess their myelination activity, i.e. by analyzing mRNA/protein levels of MBP, PLP or DM20 in isolated oligodendrocytes.
2. Figure 4E; Fig. 5; Supplementary Figure S4, : The authors state that "We used BMDMs as they share many characteristics with microglia, but they can be cultured from adult ASMko mice when SM accumulation is evident unlike microglia (Ledesma et al, 2011)" This explanation regarding the use of BMDMs is not satisfying. Adult microglia from different mouse models can be isolated and cultured for at least 24 hours (see e.g. Trias et al., 2013 *Front Cell Neurosci.* 2013 Dec 24;7:274; Wagner et al., 2017, *Acta Neuropathol Commun.* 2017 Jun 24;5(1):52 or Bornstein et al., 2013 *J Vis Exp.* 2013; (78): 50647). Furthermore, the cited reference does not discuss macrophages or microglia of ASMko mice. Thus, confirmation of the BMDM data by using adult microglia (and perhaps adult astrocytes or oligodendrocytes as negative control) would increase the impact of these findings.

Minor points.

1. We suggest to change 'lysosomal storage disorders' in the title into 'Niemann Pick disease', since this is what the data can corroborate.
2. Fig 1A: please provide a magnified image for visualization of microglial morphology.
3. Fig 2A, the staining seems very weak and with a high background. Also please provide a magnified image.
4. Fig 3E, only shows two groups although there are 6 groups in the legend.
5. Fig 4F,G: please provide a magnified image.
6. Fig 5C,J: please provide a magnified image.
7. Fig. 5E, please show the entire Ponceau S blot.
8. Fig 5G&H, please provide a housekeeping protein/Ponceau S for these blots as well.
9. Fig. 5J, an overlay of the two channels will help to appreciate the co-localization.
10. Fig 7H,I: please provide a magnified image.
11. Fig. 7a, why are there no Iba1 positive cells visible in the wt cerebellum?
12. Fig. S5, why is there no double stain of the NPA patient for Iba1 and CathB, like there is for the

NPC patient? It is impossible to tell from this chromogenic stain whether the CathB shown is found in microglia or neurons.

13. Description of osmotic mini-pump on page 11: it appears from the sentence that the Ca074Me was released s.c., shouldn't that be i.c.v?

1st Revision - authors' response

7th September 2018

EMBOJ-2018-99553

POINT-BY-POINT ANSWER TO REVIEWERS

Referee #1:

We thank this reviewer for considering our study of great interest and high quality.

I am not fully convinced that the release of CathepsinB from microglia is directly neurotoxic. There is evidence that lysosomal damage leads to release of CathepsinB into the cytosol of phagocytes, where it can activate the inflammasome pathway (Düwell et al. Nature 2010), which in turn could mediate the neurotoxic effects for example via IL1b. The authors should discuss this possibility in the discussion.

The reviewer is right Cathepsin B can cause toxicity directly or indirectly through the activation of the inflammasome pathway. However, we would like to stress that in our toxicity experiments we added the non-membrane permeable CathepsinB inhibitor Ca074 to the media after collecting it from the macrophage cultures, which were never directly exposed to the inhibitor. Therefore, our experimental setting does not allow counteracting the inflammasome response of the macrophages and targets specifically the CathepsinB that has been already secreted. Nonetheless, and to better address the interesting point raised by this reviewer, we have measured the level of IL1b in the brains of ASMko and wt mice. The finding that the level of this main component of the inflammasome response is reduced, and not increased, in the ASMko mice would argue against the activation of the pathway in the ASMko conditions. As requested by the reviewer this is now discussed in the highlighted text in page 17 and the results shown in the new Supplementary Figure S10.

The authors classify the microglia into pro- and anti-inflammatory cells based on single markers. If the authors want to keep this nomenclature more work is necessary. The authors would need to use additional markers and in particular markers that indicate pro- and anti-inflammatory changes. Alternatively, the cells could be isolated and analyzed by RNA-Seq. However, as these pro- and anti-inflammatory pathways are not part of this study, I recommend avoiding these labels.

We agree with the reviewer that the detailed pro and anti-inflammatory pathways are not the focus of this study. Following his/her recommendation we have avoided these labels throughout the text referring now to different microglia phenotypes/populations and to Arg-1 or CD16/CD32 positive microglia instead of pro and anti-inflammatory microglia.

The authors refer to sphingomyelin as a lipid enriched in myelin. In spite of its name, sphingomyelin is depleted from myelin.

It is true that, although having a key role in myelin integrity and stability, sphingomyelin is a relatively minor myelin component compared to cholesterol or cerebroside. We apologize for this misleading sentence in which we have deleted the term “enriched” to address this reviewer comment.

Page9: "We found diffuse CathB staining in a significant percentage of Iba1 positive cells in the Cb from ASMko but not wt mice, which supports LMP in ASMko microglia (Figure 4 F, H)". Additional assays are necessary to prove this point. Increased labeling is easy to see, but the diffuse labeling is not sufficient to prove lysosomal

membrane rupture. The authors could analyze lysosomal pH or determine biochemically the cytosolic fraction of CathepsinB or use other assays.

To address this reviewer query we have analysed CathepsinB levels in the cellular and cytosolic fractions of ASMko and wt BMDMs extracted with digitonin. Consistent with LMP we found increased CathB in the cytosol of ASMko BMDMs. Moreover, we have cultured microglia derived from wt and ASMko mice and analysed lysosomal integrity by lysotracker staining finding a diffuse pattern also consistent with LMP. These results are described in the highlighted text in pages 9-10 and shown in the new Figure 4F and the new Supplementary Figure S5.

The authors should in addition, test whether microglia become TUNEL positive.

To analyse microglia death we have used both cleaved Caspase-3 staining and TUNEL assays in combination with microglia identification by F4/80 labelling in the cerebellum of ASMko and wt mice. While we find increased apoptosis in the total cell population of the ASMko mice, we do not detect a significant increase neither in the number of Cleaved-Caspase-3 nor in TUNEL positive F4/80 cells. This suggests that microglia do not undergo apoptosis in the disease. We show these results in the new Supplementary Figure S9 and discuss them in the highlighted text in page 17.

Release of Cathepsin into the cytosol is known to trigger pyroptosis. The authors ablate microglia and show that this leads to worsening of the disease. Ablation of cells leads to release of potentially cytotoxic cell debris. This possibility should be discussed as a reason for the deterioration of the disease.

We thank this reviewer for this interesting comment as indeed pyroptosis and cytotoxic cell debris might worsen the disease. While we cannot completely rule out these possibilities several observations would argue against them. On one hand, we have analysed the levels of IL-1b, whose release is directly linked with pyroptosis, finding a reduction and not an increase in the brain of ASMko mice (new Supplementary Figure S10). On the other hand, it has been described that ablation of microglia by the CSF1R inhibitor PLX occurs by apoptosis very rapidly (3-5 days) after treatment initiation and have no toxic effects in wt mice. We confirmed the lack of toxicity in the wt mice after 2-month long PLX treatment. This makes unlikely that cell debris released from microglia ablation, which would appear in both wt and ASMko mice at very early stages of the treatment, is deleterious only in the ASMko mice. We discuss these issues in the highlighted text in pages 14-15 and 17.

Referee #2:

We thank this reviewer for his/her kind words on the clarity and interpretation of our results.

In the introduction the author should refer to the uniqueness of microglia origin (Ginhoux et al 2010).

We now refer to the uniqueness of microglia origin in the highlighted text in page 4.

Following on this, how the authors can formally exclude that the cells that they are looking are monocyte-derived cells that colonize the brain and not embryonic derived microglia that could die due to the KO.

The reviewer is right we cannot exclude the presence of monocyte-derived cells in the brain of ASMko mice. To address this issue we have performed staining against the transmembrane receptor TMEM119, which preferentially labels microglia and not

macrophages. We indeed see that a percentage of the F4/80 positive cells do not express TMEM119 or express it at low levels supporting their macrophage identity (but see answer to the next point on DAM microglia). The data we obtained in cultured ASMko BMDMs and the new analysis performed in primary cultured ASMko microglia confirm that both cell types show lysosomal membrane permeabilisation and Cathepsin B release. Therefore, they would similarly contribute to neuronal death irrespective of their origin. These results are shown and discussed in the highlighted text in pages 10 and 15 and in the new Supplementary Figure S6.

In the discussion the authors could discuss about the recent description of the disease-associated microglia, also called DAM, performed by Keren-Shaul et al 2017 and how this could be related to the microglia they described in this article.

We thank this reviewer for this interesting remark. It could certainly be that the protective disease-associated microglia (DAM), which have been recently described in mouse models for Alzheimer's disease and amyotrophic lateral sclerosis, also play a role in Niemann Pick diseases. Even more so as these cells upregulate Cathepsin B. We have tried to identify this microglia population in the ASMko mouse brain using antibodies against the DAM-enriched protein Trem2. However, the slight differences in intensity we have found do not allow establishing solid conclusions on the presence of DAM microglia that would require transcriptional single cell sorting. Interestingly, the expression of the microglia marker TMEM119 is downregulated in DAM microglia. It might be that the F4/80 positive cells showing none or low TMEM119 expression in the ASMko mouse brains are not infiltrated macrophages (see above point for this reviewer) but DAM microglia. This would be in agreement with very recent reports describing the absence of infiltrating peripheral myeloid cells in the brains of different mouse models for LSDs including NPC (Cho et al., 2018; Cougnoux et al., 2018). Addressing the reviewer query we now discuss these issues in pages 10 and 15 and show the results in the new Supplementary Figure S6.

Referee #3:

We thank this reviewer for acknowledging the high interest of the topic of our study.

Major points. 1. Supplementary Figure S1: To exclusively link the demyelination process to the microglia, one would need to not only count the number of oligodendrocytes, but rather assess their myelination activity, i.e. by analyzing mRNA/protein levels of MBP, PLP or DM20 in isolated oligodendrocytes.

We do not intend to exclusively link the demyelination process to the microglia. In fact, we propose that demyelination is the cause, and not the consequence, of microglia alterations since myelin debris overwhelm the phagocytic capacity of these cells damaging their lysosomes and promoting the subsequent release of toxic Cathepsin B. While it is not the scope of our study we agree with this reviewer on the relevance of understanding the reason why demyelination occurs. Buccina et al., (J Neurochemistry 2009) already analysed mRNA and expression levels of the myelin-specific proteins MBP, MAG, CNP and PLP –DM20 at birth and at early postnatal stages in ASMko mice. They found reduced mRNA and protein levels for all of them at 4 and 10 weeks of age but not at birth, when transcription factors involved in oligodendrocyte (OL) development were also found unaltered. These results favoured the hypothesis of altered myelin maintenance rather than deficient OL generation or development in ASMko conditions. This needs further investigation but to address this reviewer query we have

assessed the myelination capacity of OLs by co-staining of MBP and the axonal marker Neurofilament 200 in wt and ASMko mice treated or not with the microglia ablating drug PLX5622. The number of axons not wrapped by myelin in ASMko mice is increased compared to wt. This feature is not affected by PLX5622 treatment. This suggests that the myelination capacity of ASMko OLs is impaired but not influenced by microglia. These results are now shown and discussed in the new Supplementary Figure S2 and in the highlighted text in page 8.

2. Figure 4E; Fig. 5; Supplementary Figure S4, : The authors state that "We used BMDMs as they share many characteristics with microglia, but they can be cultured from adult ASMko mice when SM accumulation is evident unlike microglia (Ledesma et al, 2011)" This explanation regarding the use of BMDMs is not satisfying. Adult microglia from different mouse models can be isolated and cultured for at least 24 hours (see e.g. Trias et al., 2013 Front Cell Neurosci. 2013 Dec 24;7:274; Wagner et al., 2017, Acta Neuropathol Commun. 2017 Jun 24;5(1):52 or Bornstein et al., 2013 J Vis Exp. 2013; (78): 50647). Furthermore, the cited reference does not discuss macrophages or microglia of ASMko mice. Thus, confirmation of the BMDM data by using adult microglia (and perhaps adult astrocytes or oligodendrocytes as negative control) would increase the impact of these findings.

We apologize for the mistake in the reference citation that we have now corrected. Following this reviewer suggestion we have tried hard to culture adult microglia from ASMko mouse whole brain.

Unfortunately, and in agreement with the previous reports, the cellular yield was quite low. Due to the high number of cells required to detect Cathepsin B in the media we have not been able to obtain results from those cultures. In order to satisfy this reviewer query we have set up two alternative experimental approaches; a) post-natal derived microglia cultures from wt and ASMko mice in which we have confirmed the accumulation of SM after 14 days in culture by Lysenin staining. Similar to the ASMko BMDMs, these cells showed diffuse cytosolic staining of Lysotracker and the presence of extracellular cathepsin B in the culture media; b) organotypic cerebellar slice cultures from adult ASMko mice fed with PLX5622 to ablate microglia or with vehicle diet. We have analysed the presence of Cathepsin B in the media of the slice cultures finding an increase in the vehicle treated but not in the PLX treated ASMko mice. These results strongly support adult ASMko microglia as a source of secreted Cathepsin B and are described in the highlighted text in page 12 and in the new Figures 5N,O,P and in the new Supplementary Figure S5.

Minor points.

1. *We suggest to change 'lysosomal storage disorders' in the title into 'Niemann Pick disease', since this is what the data can corroborate.*

We will be willing to change the title if the referee considers it necessary. However, we would like to remark that, although with similar names, Niemann Pick diseases type A and C are two completely different LSDs caused by different genetic defects. Still, we find a similar phenotype induced by demyelination and lysosomal lipid accumulation, which are characteristics shared by many other LSDs. This moves us to propose that this is a general pathological mechanism in LSDs and to kindly ask this referee to allow us reflecting this in the title.

2. *Fig 1A: please provide a magnified image for visualization of microglial morphology.*

A magnified image of microglia morphology is now included

3. *Fig 2A, the staining seems very weak and with a high background. Also please provide a magnified image.*

We have improved the staining and provide a magnified image

4. *Fig 3E, only shows two groups although there are 6 groups in the legend.*

The graph in Fig 3E included data from all 6 groups but the merge of the lines did not allow distinguishing them. We have modified the graph to clearly show all groups.

5. *Fig 4F,G: please provide a magnified image.*

We provide a magnified image

6. *Fig 5C,J: please provide a magnified image.*

We provide a magnified image in Fig 5J. In Figure 5C there was already a magnified image. We apologize for not having reflected this in the legend. We have now corrected the text to make this point clear

7. *Fig. 5E, please show the entire Ponceau S blot.*

We show the entire Ponceau S blot in the new Supplementary figure S11

8. *Fig 5G&H, please provide a housekeeping protein/Ponceau S for these blots as well.*

The Western blots in Figures 5G and H correspond to the secreted exosome fraction and the exosome depleted culture media, respectively, from wt and ASMko BMDMs. These fractions contained too little protein to be detected by Ponceau S and we are not aware of a housekeeping protein that we could use as loading control for the culture media. However, we would like to note that the experimental setting we use provides with internal controls for protein amount since we

cultured the same number of cells in all conditions and since the exosome enriched and depleted fractions come from the very same culture media sample in each case.

9. Fig. 5J, an overlay of the two channels will help to appreciate the co-localization.

We now provide the overlay of the two channels.

10. Fig 7H,I: please provide a magnified image.

We provide a magnified image.

11. Fig. 7a, why are there no Iba1 positive cells visible in the wt cerebellum?

We apologize for the poor quality of the original image. We now provide a new one that better reflects the iba1 positive cells in the wt cerebellum.

12. Fig. S5, why is there no double stain of the NPA patient for Iba1 and CathB, like there is for the NPC patient? It is impossible to tell from this chromogenic stain whether the CathB shown is found in microglia or neurons.

We thank the referee for this comment. We now include double stain images of the NPA patient for iba1 and CathB that confirm the identity of the CathB positive cells as microglia (new Supplementary Figure S8).

13. Description of osmotic mini-pump on page 11: it appears from the sentence that the Ca074Me was released s.c., shouldn't that be i.c.v.?

The description of the osmotic mini-pump has been corrected to clarify that the pumps were implanted subcutaneously but were connected to the brain through a catheter to permit sustained Ca074Me release into the cerebrospinal fluid (highlighted text in pages 12 and 24).

2nd Editorial Decision

5th October 2018

Thank you for submitting a revised version of your manuscript. It has now been seen by two of the original referees and we have now received their comments, which are enclosed below for your information.

As you can see, they both find that all criticisms have been sufficiently addressed and recommend the manuscript for publication. However, before we can officially accept your manuscript there are a few editorial issues concerning text and figures that I would ask you to address in a final revised version.

REFeree REPORTS

Referee #1:

The paper is in my opinion now ready for publication

Referee #3:

Major points

1. The question whether the absence of microglia influences the demyelination process has been convincingly addressed in Fig. S2.
2. The analysis of Cathepsin B levels and its proform in organotypic cerebellar slice cultures and

microglia of neonatal mice supports nicely the data obtained with BMDM. Significance was not reached for Cathepsin B in the slice culture and for Pro-Cathepsin B in the neonatal microglia supernatant, which could be due to the low number of samples analyzed or different ratios of autocatalytic processing depending on the model system. Thus, the text on p 12 "We found increased CathB levels in the cerebellar tissue and in the secreted media of ASMko compared to wt mice. CathB increase was prevented by the microglia depleting treatment with PLX." should be altered by differentiating between Pro-CathB and CathB.

Minor points

Our questions have been addressed satisfactory and we do not insist on a change of the title.

2nd Revision - authors' response

12th October 2018

Please find attached the final revised version of our manuscript entitled "Lipid induced lysosomal damage after demyelination corrupts microglia protective function in Lysosomal Storage Disorders" (EMBOJ-2018-99553) addressing the remaining editorial issues as follows:

Accepted

25th October 2018

I am pleased to inform you that your manuscript has been accepted for publication in the EMBO Journal.

Corresponding Author Name: María Dolores Ledesma & Enrique Gabandé Rodríguez

Journal Submitted to: EMBO JOURNAL

Manuscript Number: EMBOJ-2018-99553